# Coiled-coil structure of meiosis protein TEX12 and conformational regulation by its C-terminal tip

James M. Dunce[1,3], Lucy J. Salmon[1] & Owen R. Davies [ID] [1,2 ✉]

Meiosis protein TEX12 is an essential component of the synaptonemal complex (SC), which mediates homologous chromosome synapsis. It is also recruited to centrosomes in meiosis, and aberrantly in certain cancers, leading to centrosome dysfunction. Within the SC, TEX12 forms an intertwined complex with SYCE2 that undergoes fibrous assembly, driven by TEX12's C-terminal tip. However, we hitherto lack structural information regarding SYCE2-independent functions of TEX12. Here, we report X-ray crystal structures of TEX12 mutants in three distinct conformations, and utilise solution light and X-ray scattering to determine its wild-type dimeric four-helical coiled-coil structure. TEX12 undergoes conformational change upon C-terminal tip mutations, indicating that the sequence responsible for driving SYCE2-TEX12 assembly within the SC also controls the oligomeric state and conformation of isolated TEX12. Our findings provide the structural basis for SYCE2-independent roles of TEX12, including the possible regulation of SC assembly, and its known functions in meiotic centrosomes and cancer.

[1] Biosciences Institute, Faculty of Medical Sciences, Newcastle University, Framlington Place, Newcastle upon Tyne NE2 4HH, UK. [2] Wellcome Centre for Cell Biology, Institute of Cell Biology, University of Edinburgh, Michael Swann Building, Max Born Crescent, Edinburgh EH9 3BF, UK. [3] Present address: Department of Biochemistry, University of Cambridge, 80 Tennis Court Road, Old Addenbrookes Site, Cambridge CB2 1GA, UK. ✉email: Owen.Davies@ed.ac.uk

Meiosis is a specialised form of cell division in which homologous chromosomes synapse, exchange genetic material by crossing over, and then segregate to generate haploid germ cells[1]. These unique processes are mediated by meiosis-specific proteins, which have diverse functions, including meiotic recombination and chromosome synapsis[2]. However, the presence of this 'toolbox' of meiotic factors within the genome poses a risk of their reactivation, unregulated function, and genomic instability in somatic cells[3,4]. Indeed, aberrantly expressed meiosis proteins are frequently expressed in cancers and can promote oncogenic processes, and thus have been described as cancer-testis antigens[5–7]. The cancer-testis antigen TEX12 is a coiled-coil protein that functions in chromosome synapsis and centrosome structure in meiosis, and can lead to centrosome dysfunction in cancers (Fig. 1a)[8,9].

The synaptonemal complex (SC) is a 'zipper'-like protein assembly that mediates synapsis of homologous chromosomes, facilitating their recombination and crossover formation, during meiosis[10]. The supramolecular SC is generated by the interactions and self-assembly of its coiled-coil protein constituents[11]. In the mammalian SC, TEX12 is an essential component that binds to SYCE2[8,12], forming a complex that self-assembles to provide the SC's fibrous 'backbone' (Fig. 1a)[13,14]. Human SYCE2-TEX12 is a rod-like 2:2 complex in which linear chains of TEX12's structural core (amino-acids 49–123) bind to an SYCE2 dimer in an intertwined coiled-coil configuration (Fig. 1b)[14]. This 'building-block' structure undergoes hierarchical assembly driven by TEX12's C-terminal tip (Ctip; amino-acids 114–123) (Fig. 1b, c)[14]. Firstly, 2:2 complexes interact in parallel, forming a 4:4 complex that is stabilised by Ctip coiled-coils binding together

interacting complexes at both ends of the molecule. A conformational change of the Ctip coiled-coiled from parallel to anti-parallel then forms end-on interactions between 4:4 complexes, which are recursive, so mediate the assembly of long fibres. Assembly of 2:2 complexes is blocked by deletion of TEX12's Ctip (ΔCtip) or glutamate mutations of its heptad amino-acids (LFIL; L110E F114E I117E L121E), whereas 4:4 complexes form but fibrous assembly is blocked by alanine mutations of solvent-exposed amino acids (FFV; F102A F109A V116A) (Fig. 1b, c)[14]. Thus, SYCE2-TEX12 can be locked into specific assembly intermediates by deletion and mutations of TEX12's Ctip.

It was recently found that TEX12 is also recruited to centrosomes, independently of SYCE2, at the same stage of meiosis (Fig. 1a)[9]. Further, TEX12 is aberrantly expressed and recruited to centrosomes in certain cancers[9]. This is associated with a poor prognosis, attributed to centrosome dysfunction that results from TEX12 recruitment[9]. These findings provide clear precedents for TEX12 having additional biological roles outside of the SC, and independently of SYCE2. Accordingly, we have reported that isolated TEX12 forms stable dimers, of high thermal stability, through the same structural core (amino-acids 49–123) that is responsible for SYCE2-binding[9]. However, it is not possible to infer the nature of the TEX12 dimer from the known SYCE2-TEX12 structures as these are intertwined folds in which there are only minimal contacts between constituent TEX12 chains[14]. Thus, to understand TEX12's centrosome function in meiosis and cancer, alongside other potential SYCE2-independent cellular roles, we sought to elucidate the structure of the TEX12 dimer.

Here, we report X-ray crystal structures of TEX12 Ctip mutants in distinct tetrameric and dimeric conformations. We

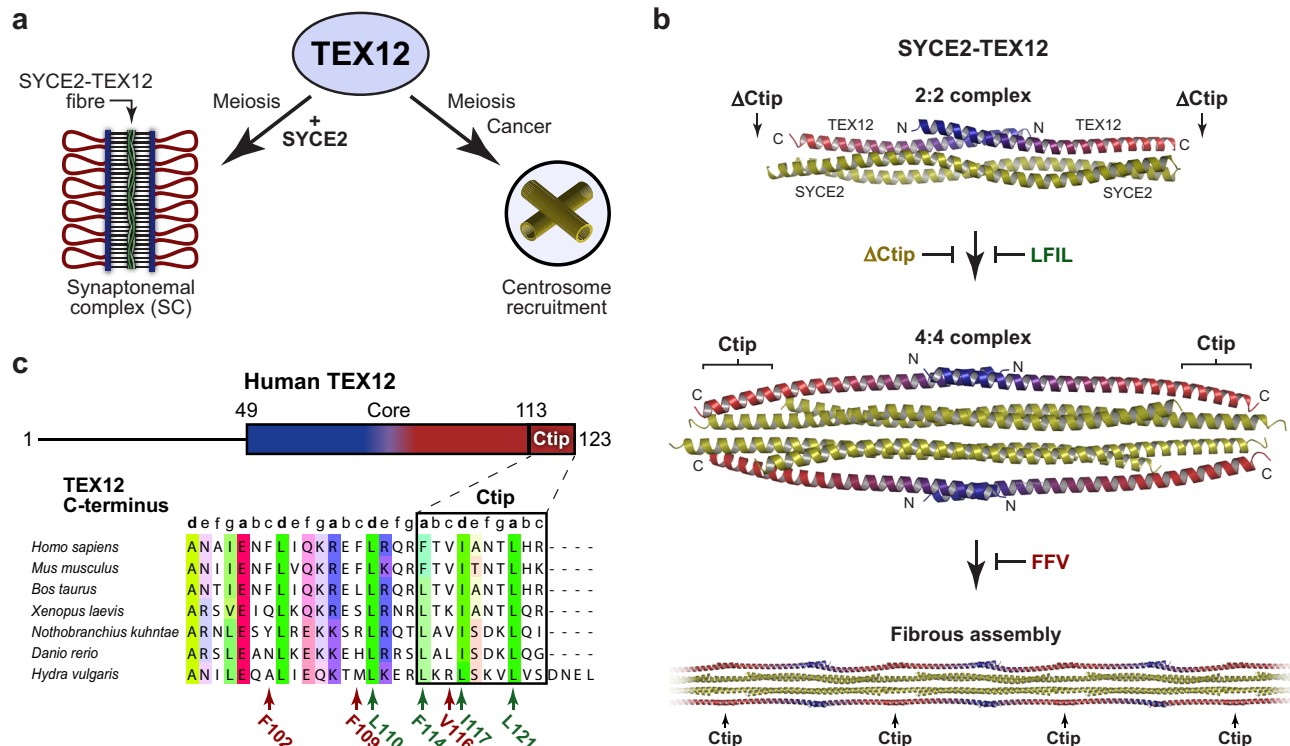

**Fig. 1 TEX12 is a synaptonemal complex and centrosomal protein. a** TEX12 has known cellular roles in meiotic chromosome synapsis and centrosome function. An SYCE2-TEX12 complex assembles into fibres to provide the backbone of the synaptonemal complex (SC) that mediates meiotic chromosome synapsis[14]. TEX12 also undergoes SYCE2-independent recruitment to centrosomes during meiosis, and in cancer cells where it contributes to centrosome dysfunction[9]. **b** In the SC, TEX12's Ctip drives hierarchical assembly of SYCE2-TEX12 from a 2:2 complex to a 4:4 complex (blocked by TEX12 ΔCtip and LFIL mutations), and into 'intermediate filament'-like fibres (blocked by TEX12 FFV mutation)[14]. **c** Schematic of the human TEX12 sequence in which its structural core (amino-acids 49–123) and Ctip (amino-acids 114–123) are highlighted, alongside a sequence alignment of TEX12's C-terminus indicating heptad predictions and amino-acids mutated in this study (modified from ref. [14]).

combine these with solution biophysics to model the wild-type TEX12 dimer, revealing a compact four-helical structure, formed by two helix–loop–helix chains, and stabilised by a Ctip coiled-coil. The diverse structures formed upon Ctip mutations reveal that Ctip regulates TEX12 oligomeric assembly in the same manner as it drives SYCE2-TEX12 assembly. Thus, we report the dimeric structure of TEX12 and how its oligomeric state is controlled by TEX12's C-terminal tip.

## Results

**TEX12's oligomeric state is regulated by its C-terminal tip.** While the structural basis of SYCE2-TEX12 assembly into the fibrous backbone of the mammalian SC has been elucidated (Fig. 1b)[14], the molecular underpinning of TEX12's recently discovered SYCE2-independent role in centrosomes remains unknown[9]. We previously demonstrated that TEX12 forms a stable homodimer[9], which is likely responsible for its SYCE2-independent functions. However, it is not possible to infer the nature of the TEX12 dimer from the SYCE2-TEX12 complex as this is an intertwined fold in which there are only minimal contacts between constituent TEX12 chains. Thus, the TEX12 dimer must represent a distinct conformation, which we sought to elucidate.

The homodimeric structure and thermal stability of human TEX12 are provided solely by its α-helical core, corresponding to amino-acids 49–123 (Fig. 1c)[9]. Hence, our structural studies focussed on this core construct, which is herein referred to as TEX12. Wild-type TEX12 was recalcitrant to crystallisation and aggregated at high concentrations (>10 mg/ml), so we wondered whether the Ctip mutations that block SYCE2-TEX12 assembly may improve its solubility and enable crystallisation (Fig. 1c). Hence, we purified TEX12 constructs harbouring a Ctip deletion (ΔCtip; amino-acids 49–113), and mutations of Ctip heptad amino-acids (LFIL; L110E F114E I117E L121E) and solvent-exposed residues (FFV; F102A F109E V116A) (Fig. 2a). These mutants remained soluble, with no overt aggregation, at high concentrations (up to 40–60 mg/ml). Circular dichroism (CD) confirmed their retention of almost entirely α-helical structure (Fig. 2b), but with a reduction in thermal stability, shown by their melting temperatures reducing from 58 °C (wild-type) to 44 °C, 30 °C and 24 °C for FFV, LFIL and ΔCtip, respectively (Fig. 2c). Thus, TEX12 Ctip mutations retain α-helical structure and increase solubility at high concentrations, but differentially reduce thermal stability.

We next used size-exclusion chromatography multi-angle light scattering (SEC-MALS) to determine the oligomeric state of TEX12 mutants. While the wild-type protein was a 17 kDa dimer, FFV formed a 33 kDa tetramer, indicating a substantial conformational change (Fig. 2d). In both cases, oligomeric species were homogeneous and stable across a 1–10 mg/ml concentration range (Fig. 2d). At 12 mg/ml, ΔCtip and LFIL mutants exhibited molecular masses of 24 and 31 kDa, respectively (Fig. 2e, f), which are intermediate between dimers and tetramers (ΔCtip—16 and 32 kDa; LFIL—18 and 36 kDa). MALS molecular masses are weight-averaged at each point in an elution profile[15]. Thus, the intermediate masses of ΔCtip and LFIL likely correspond to mixtures of dimers and tetramers, with tetramers dissociating to dimers upon dilution over SEC. Accordingly, stepwise dilution of ΔCtip and LFIL led to progressive increases in elution volume and reductions in molecular mass, down to 14 and 16 kDa dimers at 1 mg/ml (Fig. 2e, f). Thus, while TEX12 wild-type and FFV are stable dimers and tetramers, respectively, ΔCtip and LFIL form tetramers of low thermal stability that exist in concentration-dependent equilibrium with dimeric species (Fig. 2g).

**Crystal structure of a TEX12 ΔCtip anti-parallel tetramer.** The increased solubility of TEX12 ΔCtip and FFV mutants at high concentrations enabled their crystallisation and structure solution by X-ray crystallography. Crystals of TEX12 ΔCtip diffracted to 2.11 Å resolution, and we solved its structure by molecular replacement of ensembled model fragments using *AMPLE*[16,17] (Table 1 and Supplementary Fig. 1a). This revealed a rod-like tetrameric structure of 95 Å length, in which linear TEX12 chains are arranged in an anti-parallel configuration (Fig. 3a, b). The tetramer consists of two symmetry-related copies of two unique TEX12 chains that have subtle differences in side-chain positions (r.m.s. deviation = 0.58 Å). The interfaces between symmetry-related and unique chains are distinct (Fig. 3c, d), and on the basis of subsequent data, were designated as dimer and tetramer interfaces, respectively. The centre of the TEX12 ΔCtip tetramer, which have named the 'hinge' (for subsequent reasons), is characterised by reciprocal salt bridges between R78 and D82 amino-acids, in which extended arginine residues act as struts that provide increased local separation (of approximately 12 Å) between chains of the dimer interface (Fig. 3c). The hydrophobic core surrounding the hinge includes Y71 residues and is continuous with lateral four-helical bundles at both ends of the molecule (Fig. 3c, d). The four-helical bundles are stabilised by heptad interactions between chains in the dimer and tetramer interfaces, which also contribute to hydrophobic cores (Fig. 3d). Thus, the TEX12 ΔCtip crystal structure demonstrates how TEX12 forms an anti-parallel tetramer of 95 Å length in absence of its capping Ctip sequences.

**Crystal structure of a TEX12 FFV anti-parallel dimer.** We obtained two distinct crystal forms of TEX12 FFV. The first crystal form diffracted to 1.45 Å resolution, and we solved its structure by molecular replacement of ensembled model fragments using *AMPLE*[16,17] (Table 1 and Supplementary Fig. 1b). This revealed a rod-like dimer of 130 Å length in which two TEX12 chains are arranged in an anti-parallel configuration such that their Ctip sequences are located at both ends of the molecule (Fig. 4a). Similar to the dimer interface of the ΔCtip tetramer, the FFV structure contains a central 'hinge' characterised by reciprocal salt bridges between R78 and D82 amino-acids (Figs. 3c and 4b). Further, FFV's flanking anti-parallel coiled-coils demonstrate the same heptad interactions as ΔCtip's dimer interface (Figs. 3c and 4b). Accordingly, the FFV structure matches the symmetry-related dimer (r.m.s. deviation = 1.11 Å) but not the unique chain dimer (r.m.s. deviation = 4.25 Å) of the ΔCtip structure (Fig. 4c), hence our designation of the former as the dimer interface. FFV's Ctip sequences play no role in stabilising the dimeric coiled-coil, and simply emanate from either end of the molecule, where they mediate interactions within the crystal lattice. Ctip sequences arrange molecules in an end-on 'dimer-of-dimers' fashion by forming inter-molecular anti-parallel coiled-coils (Fig. 4d). Further, the hydrophobic surface of each Ctip dimer binds to the free Ctip ends of two additional 'dimer-of-dimers', forming Ctip tetramers (Fig. 4e). These Ctip interactions likely represent crystal contacts rather than biological associations. Given that TEX12 FFV is a stable tetramer in solution, its dimeric crystal structure likely represents an alternative conformation induced by crystallisation. Specifically, the extensive Ctip interactions of the crystal lattice may be mutually exclusive with interactions required for stabilisation of the solution tetramer. Hence, the crystal lattice may have disrupted the tetramer into a dimer through competitive inhibition.

**Crystal structure of TEX12 FFV in a dimeric helical assembly.** The second crystal form of TEX12 FFV diffracted to 2.29 Å

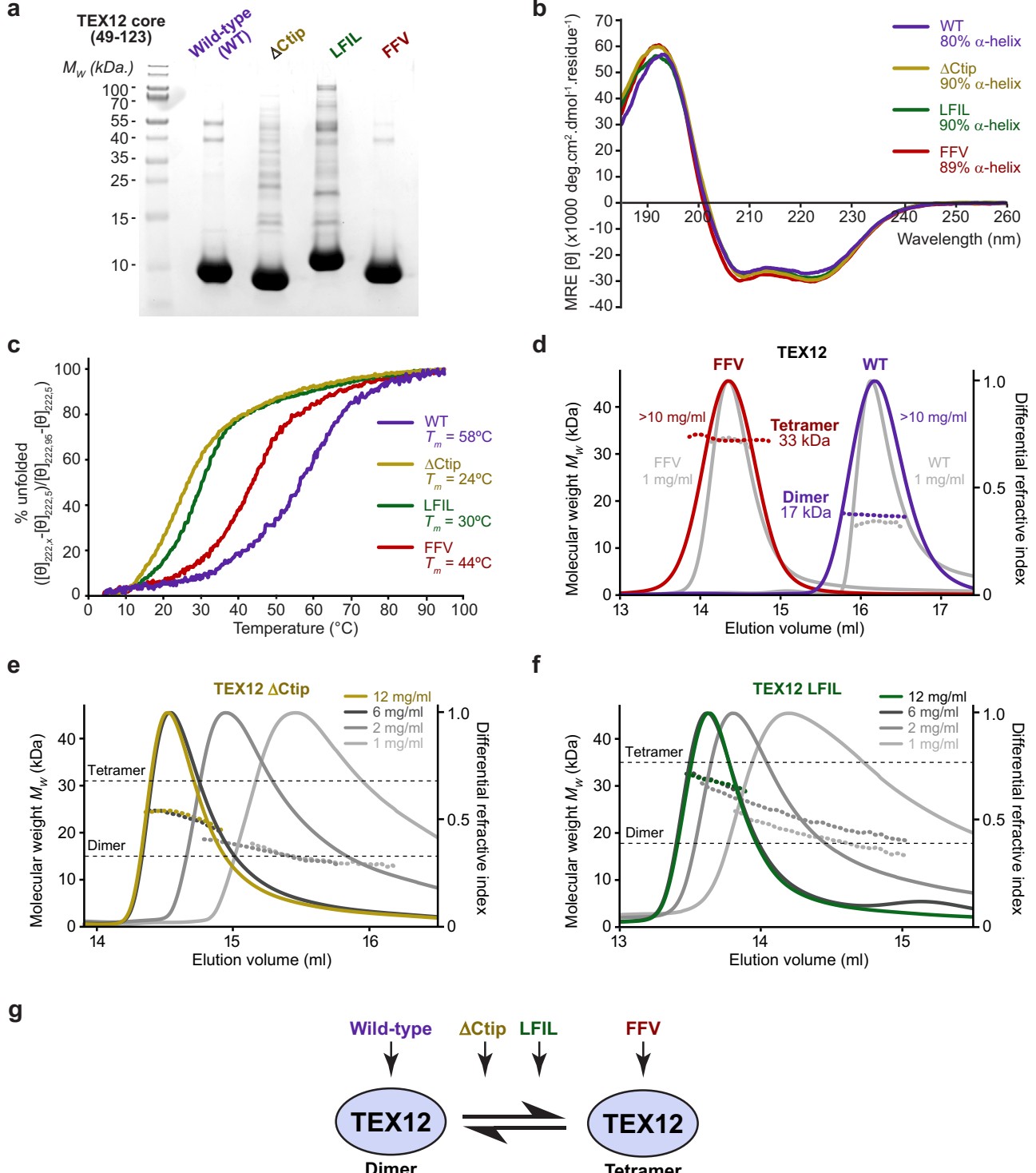

**Fig. 2 TEX12 undergoes conformational change from dimer to tetramer upon Ctip mutation. a** SDS-PAGE of purified TEX12 core (amino acids 49–123; herein referred to as TEX12) proteins containing wild-type (WT, blue), ΔCtip (deletion of amino-acids 114–123, yellow), LFIL (L110E F114E I117E L121E, green) and FFV (F102A F109E V116A, red) sequences. **b, c** Circular dichroism (CD) analysis. **b** Far UV circular dichroism (CD) spectra recorded between 260 and 185 nm in mean residue ellipticity, MRE ([θ]) (x1000 deg.cm$^2$.dmol$^{-1}$.residue$^{-1}$). Data were deconvoluted using the CDSSTR algorithm, with helical content indicated. **c** CD thermal denaturation recording the CD helical signature at 222 nm between 5 and 95 °C, as % unfolded. Melting temperatures were determined from the second derivatives of the curves. **d–f** SEC-MALS analysis in which differential refractive index (dRI; solid lines) is shown with fitted molecular weights ($M_w$; dashed lines) plotted across elution peaks. **d** TEX12 wild-type (WT) is a 17 kDa dimer, whereas the FFV mutant is a 33 kDa tetramer (theoretical—18 and 36 kDa); samples were analysed at >10 mg/ml (red and blue) and at 1 mg/ml (grey). **e, f** SEC-MALS analysis of TEX12 (**e**) ΔCtip and (**f**) LFIL mutants at concentrations between 1 and 12 mg/ml (as shown), indicating concentration-dependent dissociation of species intermediate between the theoretical masses of dimers and tetramers (ΔCtip—16 and 32 kDa; LFIL—18 and 36 kDa). **g** TEX12 is a dimer that undergoes conformational change into a tetramer by FFV mutation, and forms a mixture of less stable species, in equilibrium between dimer and tetramer, upon ΔCtip and LFIL mutations.

**Table 1 Data collection, phasing and refinement statistics.**

|  | TEX12 ΔCtip | TEX12 FFV dimer | TEX12 FFV helical |
|---|---|---|---|
| PDB accession | 6HK8 | 6HK9 | 6R2F |
| *Data collection* |  |  |  |
| Space group | P6₅22 | C222₁ | I2₁2₁2₁ |
| Cell dimensions |  |  |  |
| *a, b, c* (Å) | 47.97, 47.97, 210.98 | 43.23, 219.71, 37.50 | 59.86, 104.51, 127.51 |
| *α, β, γ* (°) | 90.00, 90.00, 120.00 | 90.00, 90.00, 90.00 | 90, 90, 90 |
| Resolution (Å) | 52.75–2.11 (2.15–2.11)[a] | 54.93–1.45 (1.48–1.45)[a] | 40.41–2.29 (2.37–2.29)[a] |
| $R_{meas}$ | 0.082 (2.330) | 0.044 (0.472) | 0.144 (1.511) |
| $R_{pim}$ | 0.014 (0.374) | 0.017 (0.170) | 0.073 (0.756) |
| $I / \sigma(I)$ | 28.1 (2.4) | 20.3 (2.6) | 8.7 (1.6) |
| $CC_{1/2}$ | 1.000 (0.890) | 0.999 (0.993) | 0.998 (0.708) |
| Completeness (%) | 92.7 (100.0) | 98.1 (96.6) | 99.6 (99.2) |
| Redundancy | 34.5 (37.5) | 7.1 (7.4) | 7.2 (7.4) |
| *Refinement* |  |  |  |
| Resolution (Å) | 41.54–2.11 | 30.97–1.45 | 40.41–2.29 |
| UCLA anisotropy (Å) | 2.2, 2.2, 2.1 | N/A | N/A |
| No. reflections | 7701 | 59176 | 18283 |
| $R_{work}/R_{free}$ | 0.2291/ 0.2580 | 0.1795/ 0.2047 | 0.2378/ 0.2621 |
| No. atoms | 1146 | 1536 | 2226 |
| Protein | 1056 | 1313 | 1991 |
| Ligand/ion | 30 | 23 | 90 |
| Water | 60 | 200 | 145 |
| *B*-factors | 55.63 | 37.67 | 54.76 |
| Protein | 54.99 | 35.55 | 54.00 |
| Ligand/ion | 73.83 | 59.85 | 60.82 |
| Water | 57.82 | 49.06 | 61.4 |
| R.m.s. deviations |  |  |  |
| Bond lengths (Å) | 0.002 | 0.008 | 0.007 |
| Bond angles (°) | 0.316 | 0.856 | 0.941 |

[a]Values in parentheses are for highest-resolution shell.

resolution, and we solved its structure by molecular replacement of ensembled model fragments using *AMPLE*[16,17] (Table 1 and Supplementary Fig. 1c). This revealed a rod-like dimer of 90 Å length in which two TEX12 chains are arranged in an anti-parallel configuration (Fig. 5a). This coiled-coil is distinct from the previous dimer and tetramer interfaces, and represents a two-heptad phase shift of its anti-parallel configuration relative to the common dimer interface of ΔCtip and FFV structures (r.m.s. deviation = 10.57 Å) (Fig. 5b, c). Owing to the phase shift, the centre of this alternative FFV dimer lacks the R78-D82 hinge present within ΔCtip and FFV dimer structures. Instead, its centre is characterised by a stacking interaction of central Y85 residues. Importantly, TEX12's Ctip sequences directly contribute to the dimeric coiled-coil, while 10 amino-acids at the TEX12's N-termini were not visible in electron density. In the crystal lattice, hydrophobic surfaces of coiled-coil ends (formed by Ctip and the N-terminus) mediate dimer-of-dimer interactions between obliquely-oriented molecules (Fig. 5d). These interactions are recursive, generating a right-handed helix of FFV dimers that is continuous throughout the lattice, and is intertwined with another FFV helix in a double-helical arrangement (Fig. 5d). These interactions likely represent crystal contacts rather than biological associations. Further, similar to the first FFV structure, these crystal lattice contacts may have induced conformational

change from a solution tetramer to a crystallographic dimer by competitively inhibiting Ctip interactions required for stability of the tetramer.

**Solution structure of the TEX12 FFV tetramer.** How do the dimeric and tetrameric crystal structures of TEX12 ΔCtip and FFV relate to the dimers and tetramers formed by TEX12 wild-type and Ctip mutants in solution? Small-angle X-ray scattering analyses protein size and shape, and can determine the length and cross-sectional radius of rod-like coiled-coil molecules in solution[14,18–20]. Hence, we analysed TEX12 wild-type and Ctip mutants by size-exclusion chromatography small-angle X-ray scattering (SEC-SAXS) (Fig. 6a, Supplementary Table 1 and Supplementary Fig. 2a).

The SAXS *P(r)* pairwise inter-atomic distance distributions of TEX12 Ctip mutants indicated maximum dimensions (*Dmax*), corresponding to coiled-coil lengths, of 110, 129 and 120 Å for ΔCtip, LFIL and FFV, respectively (Fig. 6b). These are consistent with the lengths of the ΔCtip tetramer (95 Å) and the FFV dimer (130 Å) crystal structures. Further, their cross-sectional radii were between 13 and 14 Å (Supplementary Fig. 2b). We previously demonstrated through analysis of several SC proteins in different oligomeric conformations that dimeric coiled-coils typically have SAXS cross-sectional radii of 8–9 Å, whereas four-helical have larger cross-sectional radii of 10–14 Å[14,18,20,21]. Hence, values of 13–14 Å suggest the presence of four-helical, rather than two-helical, coiled-coil structure. These findings are consistent with TEX12 Ctip mutants forming extended rod-like tetramers in solution (ΔCtip and LFIL were analysed at ~30 mg/ml to minimise dissociation). Further, the ΔCtip SAXS scattering curve was closely fitted by the ΔCtip tetrameric crystal structure upon addition of the few missing C-terminal amino-acids ($\chi^2 = 1.54$; Fig. 6a), confirming that the ΔCtip crystal structure is representative of its tetrameric structure in solution.

We hypothesised that FFV adopts the same conformation as the ΔCtip tetramer, with its flanking Ctip sequences forming stabilising dimeric coiled-coils that prevent its dissociation. To test this, we modelled the FFV tetramer by docking an idealised extended Ctip coiled-coil dimer (in which heptad residues include L110, F114, I117 and L121) onto both ends of the ΔCtip tetramer structure, utilising overlapping sequence to achieve seamless connections, followed by energy minimisation and geometry idealisation (Fig. 6d). The resultant model closely fits the FFV SAXS scattering curve ($\chi^2 = 1.94$; Fig. 6a), its 125 Å length corresponds to the SAXS *P(r)* maximum dimension of 120 Å (Fig. 6b), and its shape and size closely match the SAXS ab initio molecular envelope (Fig. 6c). Thus, we conclude that TEX12 ΔC-tip forms the anti-parallel tetramer observed in its crystal structure, albeit prone to concentration-dependent dissociation, and the same tetrameric structure is stabilised within TEX12 FFV by Ctip coiled-coils 'tying-off' both ends of the molecule.

**Solution structure of the wild-type TEX12 dimer.** In contrast with Ctip mutants, the SAXS *P(r)* distribution of the wild-type TEX12 dimer revealed a 66 Å length (Fig. 6b), which is almost half the length of the extended TEX12 chains within ΔCtip tet-ramer and FFV dimer structures. However, its cross-sectional radius of 12 Å indicates that it also has four-helical structure (Supplementary Fig. 2b). These findings can be explained by TEX12 chains folding back on themselves, into helix–loop–helix conformations, such that a four-helical core is formed from two TEX12 chains. We reasoned that the core may correspond to one-half of the FFV tetramer, with its helices connected by loops rather than being continuous chains, which we tested by building a compact dimer model.

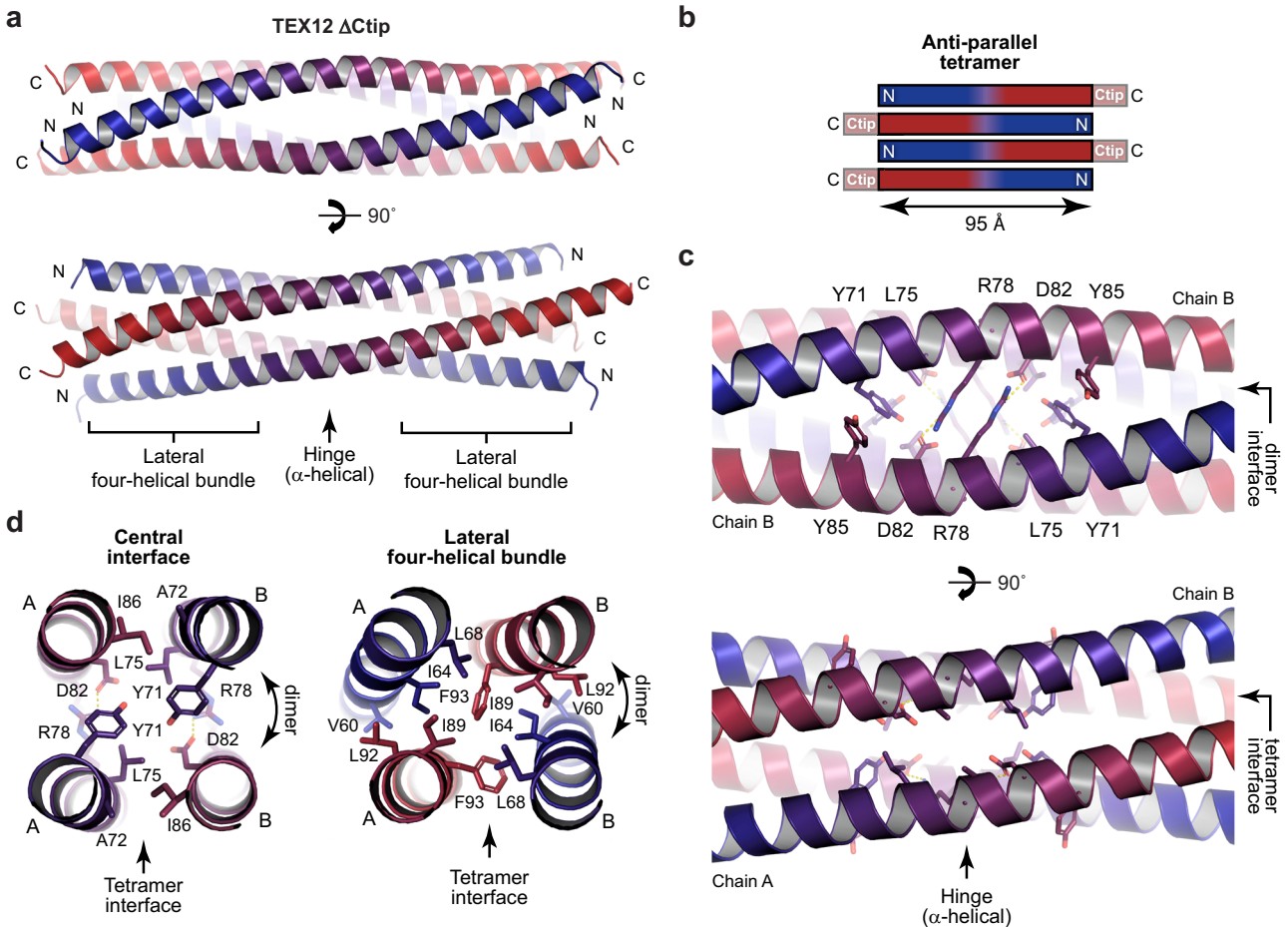

**Fig. 3 Crystal structure of a TEX12 ΔCtip tetramer. a** Crystal structure of TEX12 ΔCtip in a tetrameric conformation. The four TEX12 chains (coloured blue to red in an N- to C-terminal direction) are arranged in an anti-parallel configuration with a central 'hinge' flanked by two lateral four-helical bundles. **b** Schematic of the TEX12 ΔCtip structure highlighting the location of missing Ctip sequences in pairs on either side of the 95 Å core. **c** The hinge is defined by salt-bridges between R78 and D82 amino-acids, and is supported by surrounding aromatic packing interactions of symmetry-related chains (dimer interface), and limited hydrophobic interactions between alternative chains A and B (tetramer interface). **d** The lateral four-helical bundles contain 'coiled-coil'-like interactions between symmetry-related chains (dimer interface), and hydrophobic and aromatic interactions between opposing dimers (tetramer interface).

We previously defined the region surrounding the R78-D82 salt bridges of the ΔCtip tetramer and FFV dimer as the hinge as this lies at the centre of symmetry of these structures, and thereby of the FFV tetramer (Fig. 6d and Supplementary Fig. 3a), so is the only site where it is possible to form a compact dimer by adoption of an alternative loop conformation. Hence, we took one-half of the FFV tetramer, with chains truncated to retain the hinge's two R78-D82 salt bridges (Supplementary Fig. 3b). The N-terminal helices (harbouring R78 residues) were one turn longer than C-terminal helices (harbouring D82 residues), so to obtain a structure in which both helices terminated proximally, we extended the deletion to amino-acids 76–80 (Supplementary Fig. 3c). We then modelled these deleted amino-acids as loops between the most proximate chains, resulting in a model of the TEX12 compact dimer (Fig. 6e). As the TEX12 compact dimer model corresponds to half of the FFV tetramer, it retains a structural core consisting of almost entirely hydrophobic amino-acids, with charged amino-acids located on the surface (Supplementary Fig. 4). Further, R78-D82 salt bridges of the α-helical hinge conformation in ΔCtip and FFV structures were restored within the looped hinge of the compact dimer model (Supplementary Fig. 3d).

We tested the validity of our TEX12 compact dimer model by subjecting it to molecular dynamics simulations at 37 °C. In three replicates of 1 μs simulations, the model remained intact and

retained its hydrophobic core and R78-D82 salt bridges within the looped hinge (Fig. 7a and Supplementary Data 1). The overall r.m.s. deviation was constant throughout the runs, at values of typically between 1.5 and 2.5 Å (Fig. 7b, c and Supplementary Fig. 5). Further, local r.m.s. fluctuations were around 1 Å for the two helices (including most of the Ctip), between 1.5 and 2 Å at the looped hinge, and much larger values at the unstructured termini (Fig. 7d). Further, α-helical secondary structure was retained throughout the trajectories (Supplementary Fig. 5). As controls, we performed 100 ns simulations of models in which the sequence register was varied by 1–8 amino-acids. In these simulations, the cores ballooned almost immediately, giving overall r.m.s. deviations of between 4 and 6 Å within the first 10 ns (Supplementary Fig. 6a–c and Supplementary Data 1). Their structures further distorted over the full 100 ns, in some cases showing complete dissociation of chains, with overall r.m.s. deviations of up to 21 Å. Notably, the most severe structural distortion occurred upon register shifts of 1, 2, 5 and 8 amino-acids, in contrast with shifts of 3, 4 and 7 amino-acids that locate hydrophobic residues within the core owing to the 3.5 amino-acid periodicity of coiled-coils (Supplementary Fig. 6c). Hence, molecular dynamics simulations demonstrate the plausibility of the model in explaining the compact dimer structure of wild-type TEX12.

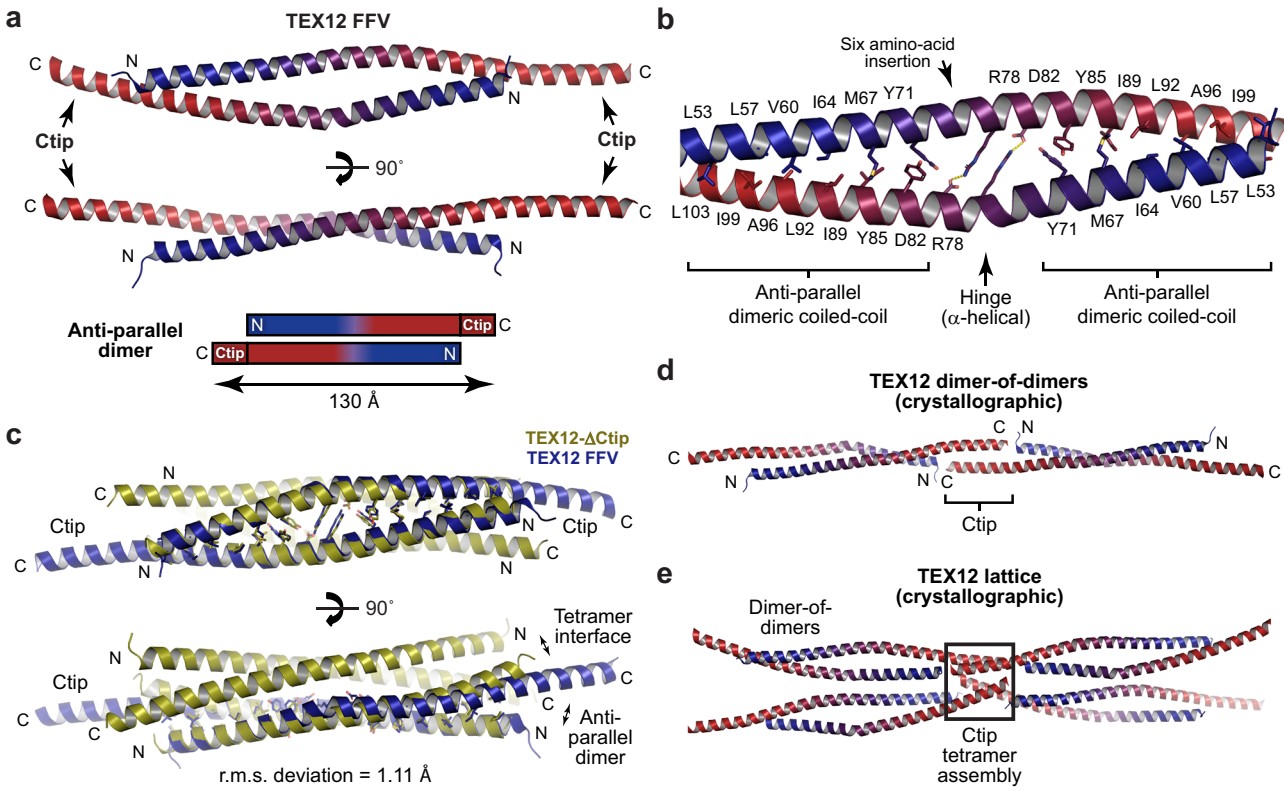

**Fig. 4 Crystal structure of TEX12 FFV in a dimeric conformation. a** Crystal structure of a TEX12 FFV dimer. The two TEX12 chains (coloured blue to red in an N- to C-terminal direction) are arranged in an anti-parallel configuration such that their Ctip sequences project from either side of the core dimer, formed of amino-acids 49–113, to form a 130 Å molecule. **b** The dimeric core is formed of two anti-parallel coiled-coils with a midline six amino-acid insertion, forming a central hinge of R78-D82 salt bridges that act as struts to locally separate the interacting chains. **c** Superposition of the TEX12 FFV dimer structure (blue) and a constituent dimer of the TEX12 ΔCtip structure (yellow), demonstrating their shared hinge and dimer interface (r.m.s. deviation of 1.11). **d, e** The TEX12 FFV crystal lattice includes (**d**) end-on dimer-of-dimers assembly through formation of anti-parallel coiled-coils between Ctip sequences, which (**e**) interact with the Ctip ends of two additional molecules within anti-parallel Ctip tetrameric assemblies.

We next assessed the compact dimer model in relation to experimental SAXS data. The model closely fits the wild-type TEX12 SAXS scattering curve ($\chi^2 = 1.15$; Fig. 6a), its 65 Å length corresponds to the SAXS $P(r)$ maximum dimension of 66 Å (Fig. 6b), and its shape and size closely match the SAXS ab initio molecular envelope (Fig. 6c). We confirmed these findings using point mutation F109E, which increased the solubility of the TEX12 dimer at high concentrations, allowing us to collect SAXS data with a higher signal-to-noise ratio. The compact dimer structure closely fits the F109E SAXS scattering curve ($\chi^2 = 1.84$), and matches the dimensions of its $P(r)$ distribution and ab initio molecular envelope (Supplementary Fig. 7a–f). Thus, the compact dimer model explains the SEC-SAXS data collected on both wild-type and F109E constructs, so likely represents the structure of the wild-type TEX12 dimer in solution.

**Model for TEX12 structure regulation by its C-terminal tip**. We can now explain the observed behaviour of TEX12 wild-type and Ctip mutants through a simple model (Fig. 8a–d). In Ctip mutants, TEX12 forms dimers through a conserved dimer interface, which associate into tetramers through a conserved tetramer interface (Fig. 8a–c). In FFV, flanking Ctip coiled-coils stabilise the tetramer by 'tying-off' both ends of the molecule, accounting for the stability of its tetramer in solution (Fig. 8a). These Ctip coiled-coil interactions are absent in ΔCtip and LFIL mutants, owing to the absence (ΔCtip) and disruption of the heptad interface (LFIL). Thus, their tetramers are substantially less stable, and exist in concentration-dependent equilibrium with

dimers (Fig. 8b, c). Wild-type TEX12 dimers contain the same dimer, tetramer and Ctip interfaces. However, integrity of amino-acids F102 and V116 confers additional stability to the Ctip coiled-coil that favours formation of a compact dimer in which a 'half-tetramer' is formed of two helix–loop–helix TEX12 chains by the R78-D82 hinge undergoing conformation change from α-helices to loops (Fig. 8d). Thus, our findings reveal the structure of the wild-type TEX12 dimer, and how its oligomeric assembly is regulated by the C-terminal tip.

## Discussion

Our crystallographic and biophysical data reveal that TEX12 adopts distinct oligomeric states and conformations upon subtle modifications of its C-terminal tip. Its apparently diverse compact dimer, extended dimer and tetramer conformations are structurally underpinned by three common TEX12 interfaces. The most fundamental unit is formed by the dimer interface, which involves anti-parallel coiled-coil interactions surrounding a central R78-D82 salt bridge hinge, and is the sole interface within the FFV dimer crystal structure. The tetramer interface, mediated by coiled-coil heptad and hydrophobic interactions, binds together constituent dimers into tetramers, as observed in the ΔCtip tetramer crystal structure. Finally, the Ctip interface, consisting of a parallel coiled-coil with heptad amino-acids L110, F114, I117 and L121 (disrupted in LFIL), stabilises the four-helical core by tying-off the molecule, at both ends in the FFV tetramer and at one end in the wild-type compact dimer.

The three conserved TEX12 interfaces explain our biophysical findings of the structure and stability of TEX12 wild-type and

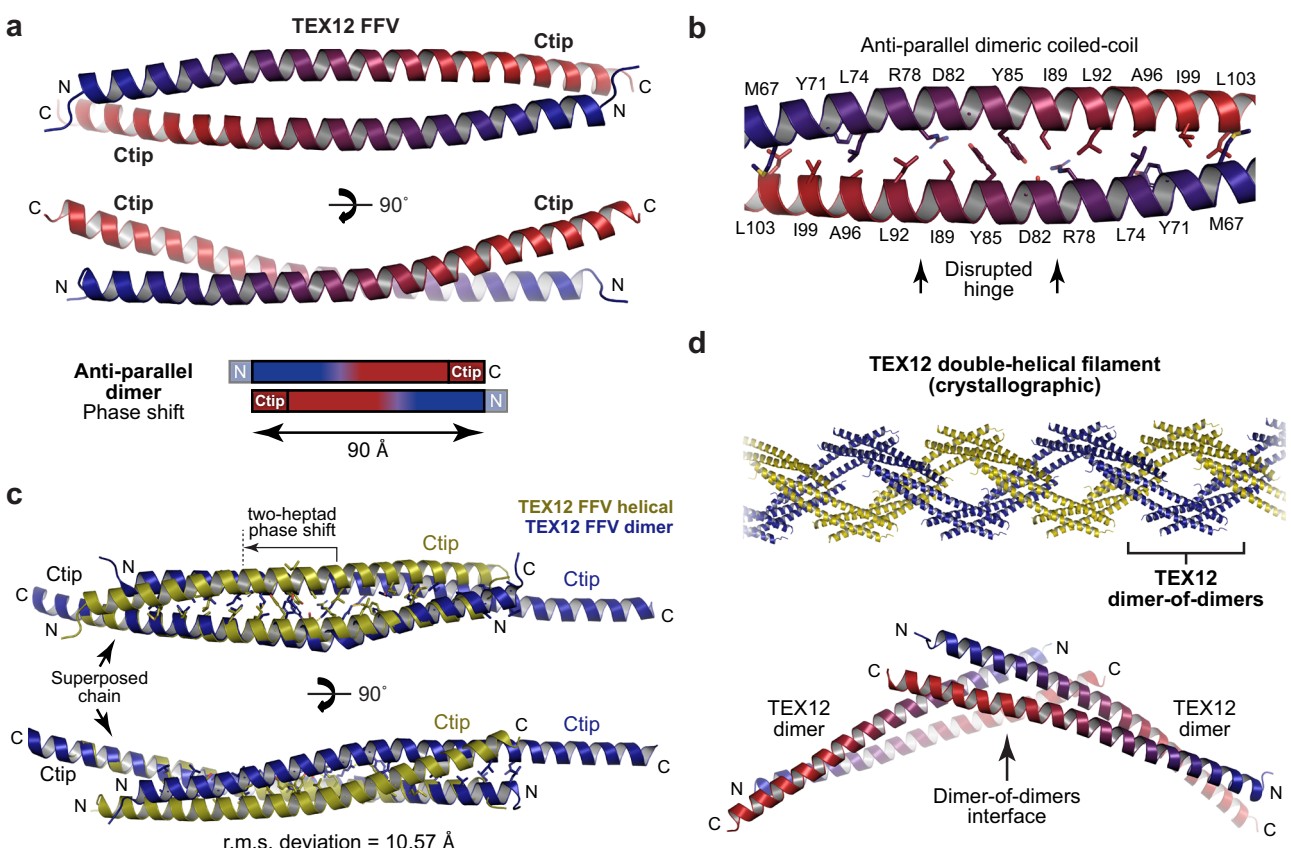

**Fig. 5 Crystal structure of TEX12 FFV in a dimeric helical conformation. a** Constituent dimer of the crystal structure of a TEX12 FFV helix. The two TEX12 chains (coloured blue to red in an N- to C-terminal direction) are arranged in an anti-parallel configuration that includes their Ctip sequences and excludes N-terminal amino-acids 49–59, providing a length of 90 Å. **b** The dimeric core is a single continuous anti-parallel coiled-coil, which contains a central aromatic interface between Y85 residues, and lacks the R78-D82 salt bridge hinge. **c** Comparison of the TEX12 FFV dimer structure in dimeric (blue) and helical (yellow) conformations in which structures were superposed based on a single chain, as indicated (r.m.s. deviation = 10.57). Their dimeric cores are distinct and defined by a two-heptad phase shift of the helical structures in comparison with the dimeric structure. **d** In the TEX12 FFV crystal lattice, individual molecules interact through a recursive dimer-of-dimer interface to form right-handed helices that are intertwined in a double-helical arrangement.

Ctip mutants in solution (Fig. 8a–d). TEX12 FFV dimers form tetramers that are stabilised by tetramer interfaces and Ctip coiled-coils, which bind together interacting dimers at both ends of the molecule (Fig. 8a). The stability afforded by all three interfaces accounts for its high thermal stability and formation of a non-dissociating tetramer in solution. TEX12 ΔCtip and LFIL dimers form tetramers through the same interfaces, but lack stabilising Ctip coiled-coils, owing to their deletion (ΔCtip) and disruption (LFIL) (Fig. 8b, c). This reduced stability accounts for their low melting temperatures and formation of dissociating tetramers and dimers in solution. In wild-type TEX12, the hinge undergoes conformational change from α-helices to loops, forming the four-helical core from two helix–loop–helix chains that are bound together by a Ctip coiled-coil (Fig. 8d). The compact dimer contains all three interfaces, with the Ctip coiled-coil supported by intact F102 and V116 amino-acids. Thus, it has greater stability than the FFV tetramer, accounting for its very high thermal stability and formation of a non-dissociating compact dimer in solution.

The stabilisation of TEX12 structures by Ctip coiled-coils bears striking resemblance to their roles in SYCE2-TEX12 assembly (Fig. 1b)[14]. In both cases, Ctip coiled-coils stabilise oligomeric assemblies by binding together interacting components at the ends of molecules. Further, their mutations have analogous consequences. Ctip and LFIL disrupt the coiled-coil, destabilising TEX12 tetramers and SYCE2-TEX12 4:4 complexes, whereas FFV

sustains these structures but prevents formation of TEX12 compact dimers and SYCE2-TEX12 fibres (Figs. 1b and 8a–d)[14]. Thus, TEX12's C-terminal tip has similar roles in controlling TEX12 structure and driving SYCE2-TEX12 assembly (Figs. 1b and 8a–d).

Our crystal structure of an FFV dimer in a helical conformation demonstrates a unique, phase-shifted anti-parallel conformation. This crystal form involves extensive lattice interactions, and its dimer and helical dimer-of-dimers structure are incompatible with our SEC-SAXS data for all TEX12 constructs. Thus, while we cannot exclude the possibility of the helical dimer forming in some context in vivo, it does not explain any solution state of TEX12, and likely constitutes an artefact of crystallisation.

Alphafold2 has revolutionized structure prediction of individual protein domains and in many cases complexes[22,23]. However, in our experience, its performance at predicting coiled-coil oligomers is limited, likely owing to their subtle sequence differences and low representation in training datasets. Alphafold2 multimer predicted an extended anti-parallel TEX12 structure that has a similar dimer interface to the FFV helical structure (r.m.s. deviation = 2.43 Å) (Supplementary Fig. 8a–d). However, it did not predict the conserved ΔCtip and FFV dimer interface that defines its native conformation (r.m.s. deviation ≈ 10 Å), and failed to predict the helix–loop–helix compact dimer structure of the wild-type protein (Supplementary Fig. 8d). This highlights the

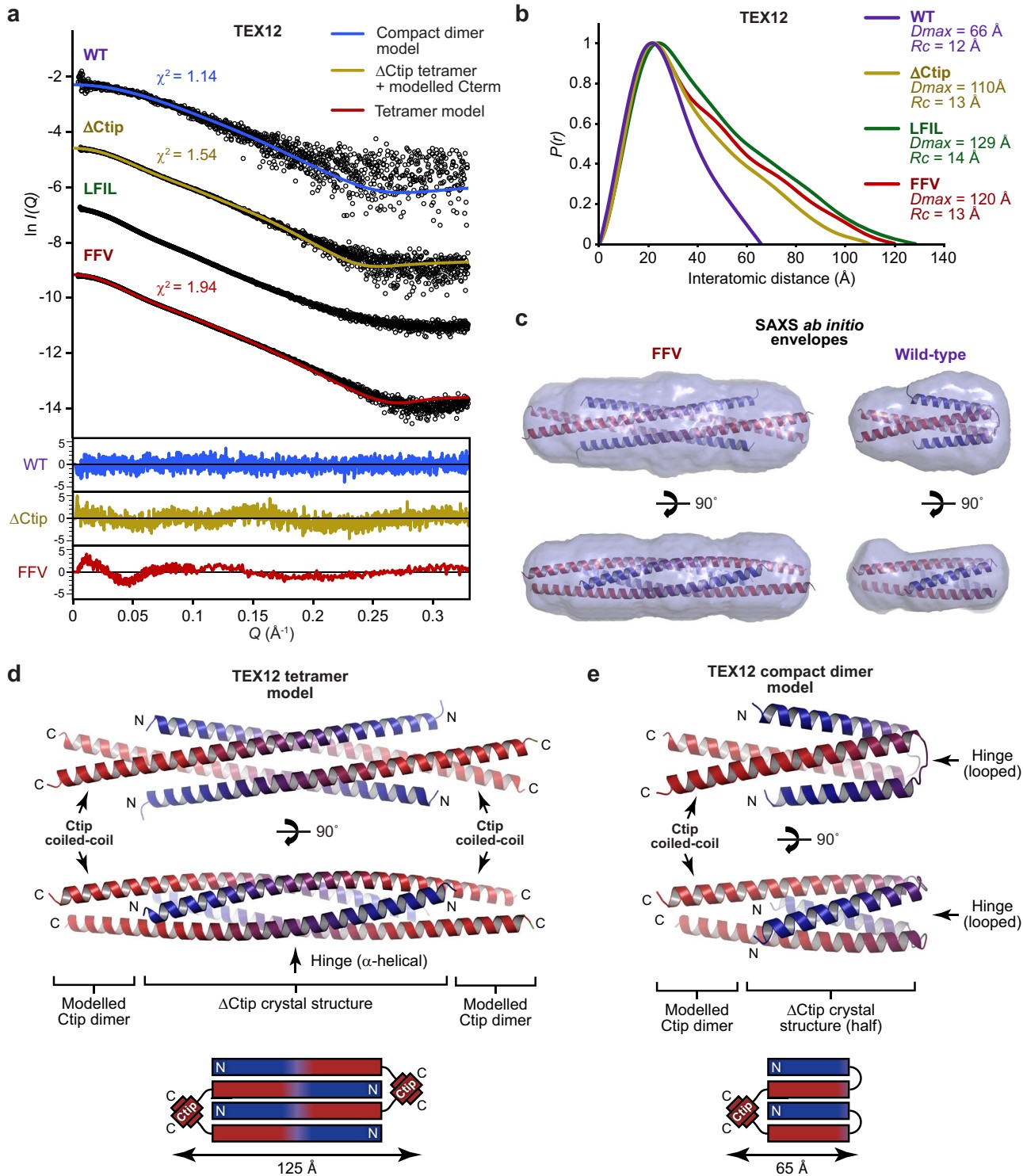

**Fig. 6 TEX12 adopts a compact dimeric conformation in solution. a–c** SEC-SAXS analysis of TEX12 core wild-type (blue), ΔCtip (yellow), LFIL (green) and FFV (red). **a** SAXS scattering data overlaid with the theoretical scattering curves of the compact dimer model, ΔCtip tetramer structure with modelled C-terminus and tetramer model for wild-type, ΔCtip and FFV, respectively; $\chi^2$ values are indicated and residuals for each fit are shown (inset). **b** SAXS $P(r)$ inter-atomic distance distributions in which maximum dimensions (*Dmax*) are indicated, along with cross-sectional radii (*Rc*) determined from Guinier analysis (Supplementary Fig. 3a, b). **c** SAXS ab initio models of FFV and wild-type TEX12. Filtered averaged models from 30 independent *DAMMIF* runs are shown with the TEX12 tetramer and compact dimer models docked into the SAXS envelopes. **d** Theoretical model of the TEX12 tetramer structure (corresponding to FFV) in which parallel dimeric coiled-coils of Ctip sequences were docked onto the core ΔCtip crystal structure to form an extended molecule of 125 Å length. **e** Theoretical model of the TEX12 compact dimer consisting of one-half of the tetramer model in which N- and C-terminal helices were joined by modelling the hinge in an alternative conformation of five amino-acid loops; the length of the theoretical compact dimer structure is 65 Å.

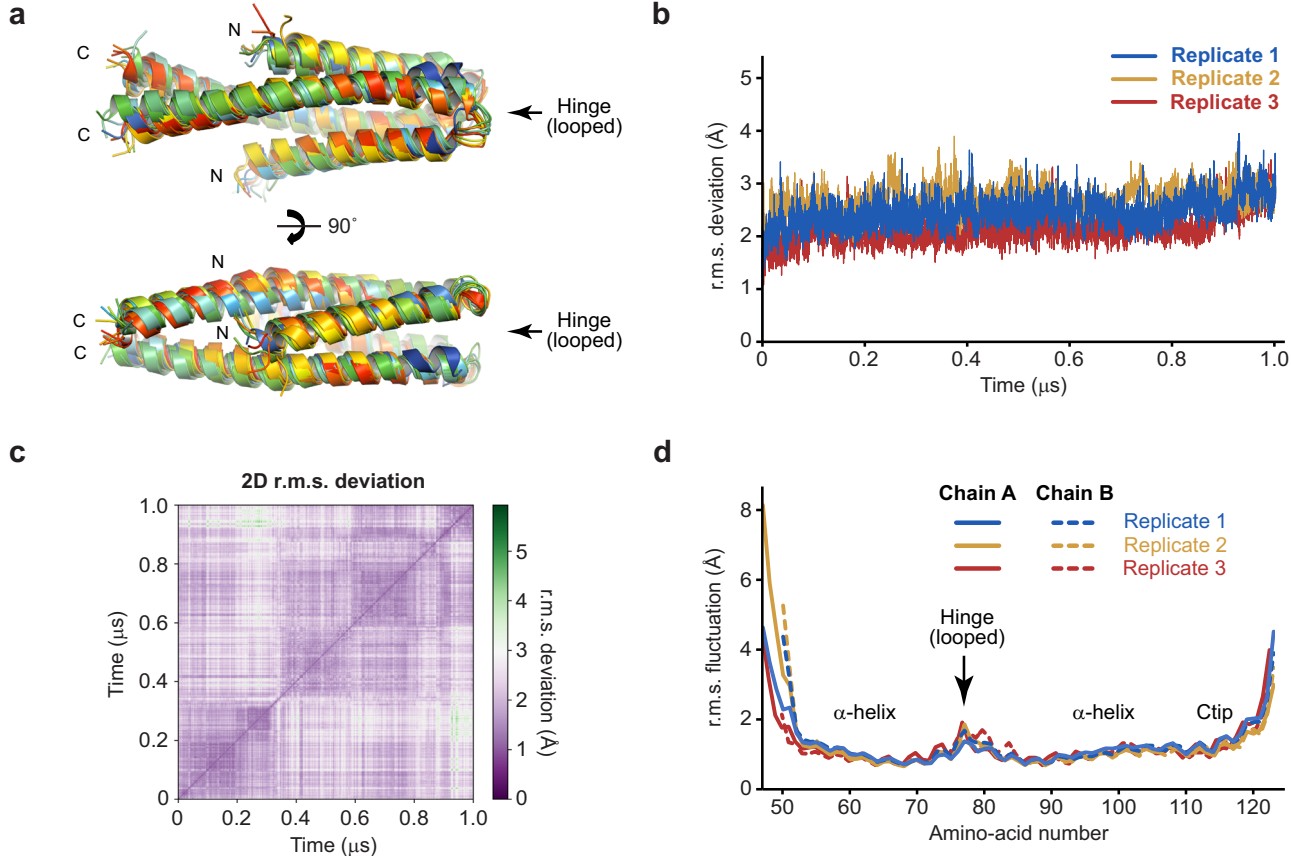

**Fig. 7 Molecular dynamics simulations of the TEX12 compact dimer structure. a–d** Analysis of the TEX12 compact dimer structure across 1-μs molecular dynamics simulations performed at 37 °C (n = 3). **a** Superimposed TEX12 compact dimer structures at 100-ns intervals of a representative trajectory, coloured from blue (0 ns) to red (1 μs). **b** Overall r.m.s. deviations and **c** 2D r.m.s. deviations (corresponding to panel **a**) across 1-μs trajectories (2D r.m.s. deviations for the remaining replicates are shown in Supplementary Fig. 6). **d** Individual amino-acid r.m.s. fluctuations following 1-μs trajectories, shown for both chains of the dimer (solid and dashed), and indicating the positions of the α-helices, looped hinge and Ctip.

ongoing importance of experimental structure elucidation and biophysical validation in the post-Alphafold2 era.

The diverse oligomeric structures formed by TEX12, in isolation and within SYCE2-TEX12 complexes, induced by subtle mutations of its C-terminal tip, highlight the potential for coiled-coil proteins to adopt an unusually large number of alternative conformations (Fig. 9). This 'promiscuity' has been observed in numerous systems[24,25], including archetypal coiled-coil GCN4, which transitions between dimers, trimers and tetramers upon point mutations[26]. The few contacts within coiled-coil interfaces, and their similarity between distinct oligomers, can result in relatively similar free-energies of folding[27,28]. Hence, minor sequence alterations can alter which oligomer has the lowest energy state and is thereby the dominant state in solution, and some sequences support multiple alternative conformations owing to their isoenergetic landscapes[29]. In vivo, conformational changes can be triggered by local chemical environments or protein interactions. Indeed, conformational switching of coiled-coils has been reported in response to pH[18,24,27], and within macromolecular protein assemblies[30]. Further, TEX12 compact dimer formation through adoption of a looped hinge conformation bears intriguing similarity to SMC-kleisin complexes MukBEF and cohesin, which transition from extended to folded conformations through loop formation at 'elbows' within their coiled-coils[31,32]. Thus, their ability to adopt multiple conformations, allows individual coiled-coil proteins to participate in diverse cellular functions and in structurally dynamic processes.

How are the biological functions of TEX12 explained by its diverse oligomeric structures? In meiosis, the TEX12 dimer may act as a storage form that competitively inhibits its interaction with SYCE2 and regulates the formation of SYCE2-TEX12 fibres during SC assembly[14]. This could dynamically restrict SYCE2-TEX12 assembly until a timely stage of synapsis, and promote disassembly upon crossover formation. The TEX12 dimer likely underpins its SYCE2-independent recruitment to meiotic centrosomes, and centrosome dysfunction in cancer[9]. Further, alternative TEX12 conformations may be induced by local conditions and/or protein-binding that affect its Ctip. Thus, our elucidation of TEX12 structure and its regulation by the C-terminal tip provide the molecular basis for ongoing functional studies of TEX12's biological roles in the SC, centrosomes and cancer. Moreover, our findings may facilitate in vivo discoveries, such as by establishing imaging methods that differentiate TEX12 dimers and SYCE2-TEX12 complexes, and in developing new diagnostic, prognostic and therapeutic methods for TEX12-positive cancers.

## Methods

**Recombinant protein expression and purification.** Sequences corresponding to human TEX12 core (amino-acids 49–123) wild-type, F102A F109E V116A (FFV), L110E F114E I117E L121E (LFIL) and ΔCtip (amino-acids 49–113) were cloned into pMAT11 vectors[33] for expression as TEV-cleavable N-terminal His-MBP-fusion proteins. Constructs were expressed in BL21 (DE3) cells (Novagen®) in 2xYT media, induced with 0.5 mM IPTG for 16 h at 25 °C. Cells were lysed by sonication in 20 mM Tris pH 8.0, 500 mM KCl, and fusion proteins were purified from clarified lysate through consecutive Ni-NTA (Qiagen), amylose (NEB) and

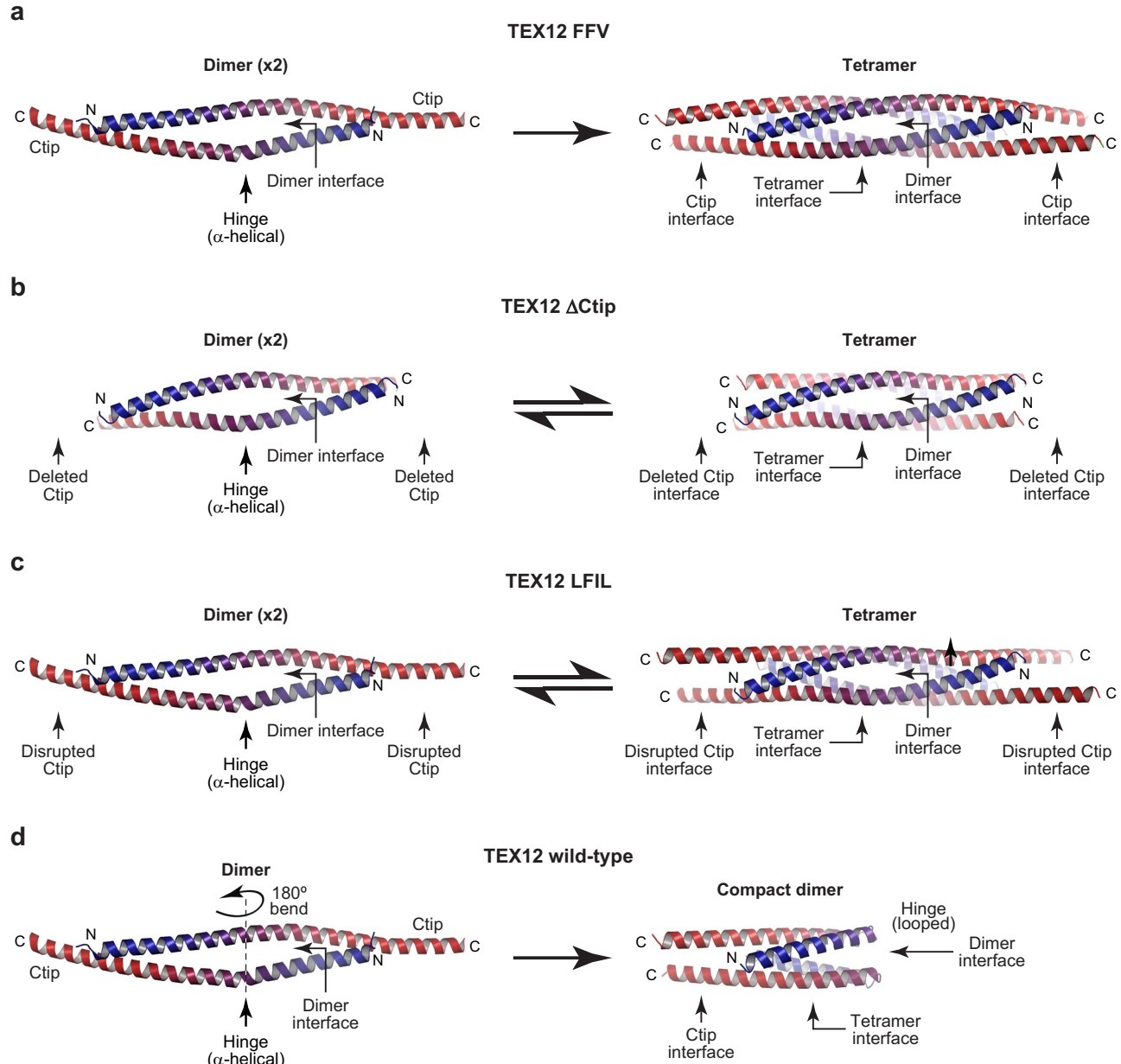

**Fig. 8 Model for the oligomeric state and conformation of TEX12 wild-type and Ctip mutants. a–d** The oligomeric state and conformation of TEX12 wild-type and Ctip mutants in solution can be explained by their conserved dimer, tetramer and Ctip interfaces. **a** TEX12 FFV dimers associate into tetramers that are stabilised by Ctip interfaces at either end of the molecule, accounting for the sole presence of extended tetramers in solution. **b** TEX12 ΔCtip and **c** TEX12 LFIL dimers associate into tetramers that remain unstable owing to their deleted and disrupted Ctip interfaces, accounting for the concentration-dependent dissociation of tetramers into mixtures of tetramers and dimers in solution. **d** Wild-type TEX12 dimers fold back on themselves by adopting helix–loop–helix structures in which the hinge transitions from α-helical to looped conformations. In this structure, the dimer and tetramer interfaces are formed by only two chains, and are stabilised by a Ctip interface supported by intact F102 and V116 amino-acids, accounting for the sole presence of compact dimers in solutions.

HiTrap Q HP (GE Healthcare) ion exchange chromatography. Affinity tags were removed by incubation with TEV protease and cleaved samples were purified by HiTrap Q HP ion exchange chromatography and size-exclusion chromatography (HiLoad 16/600 Superdex 200, GE Healthcare) in 20 mM Tris pH 8.0, 150 mM KCl, 2 mM DTT. Protein samples were concentrated using Pall 3 kDa Microsep™ Advance centrifugal devices and were stored at −80 °C following flash-freezing in liquid nitrogen. Protein samples were analysed by SDS-PAGE with Coomassie staining, and concentrations were determined by UV spectroscopy using a Cary 60 UV spectrophotometer (Agilent) with extinction coefficients and molecular weights calculated by ProtParam (http://web.expasy.org/protparam/).

**Size-exclusion chromatography multi-angle light scattering (SEC-MALS).** The absolute molecular masses of TEX12 constructs were determined by size-exclusion chromatography multi-angle light scattering (SEC-MALS). Protein samples at indicated concentrations were loaded onto a Superdex™ 200 Increase 10/300 GL size exclusion chromatography column (GE Healthcare) in 20 mM Tris pH 8.0, 150 mM KCl, 2 mM DTT, at 0.5 ml/min using an ÄKTA™ Pure (GE Healthcare). The column outlet was fed into a DAWN® HELEOS™ II MALS detector (Wyatt Technology), followed by an Optilab® T-rEX™ differential refractometer (Wyatt Technology). Light scattering and differential refractive index data were collected and analysed using ASTRA® 6 software (Wyatt Technology). Molecular weights and estimated errors were calculated across eluted peaks by extrapolation from Zimm plots using a dn/dc value of 0.1850 ml/g. SEC-MALS data are presented as differential refractive index (dRI) profiles, with fitted molecular weights ($M_W$) plotted across elution peaks.

**Circular dichroism (CD) spectroscopy.** Far UV circular dichroism (CD) spectroscopy data were collected on a Jasco J-810 spectropolarimeter (Institute for Cell

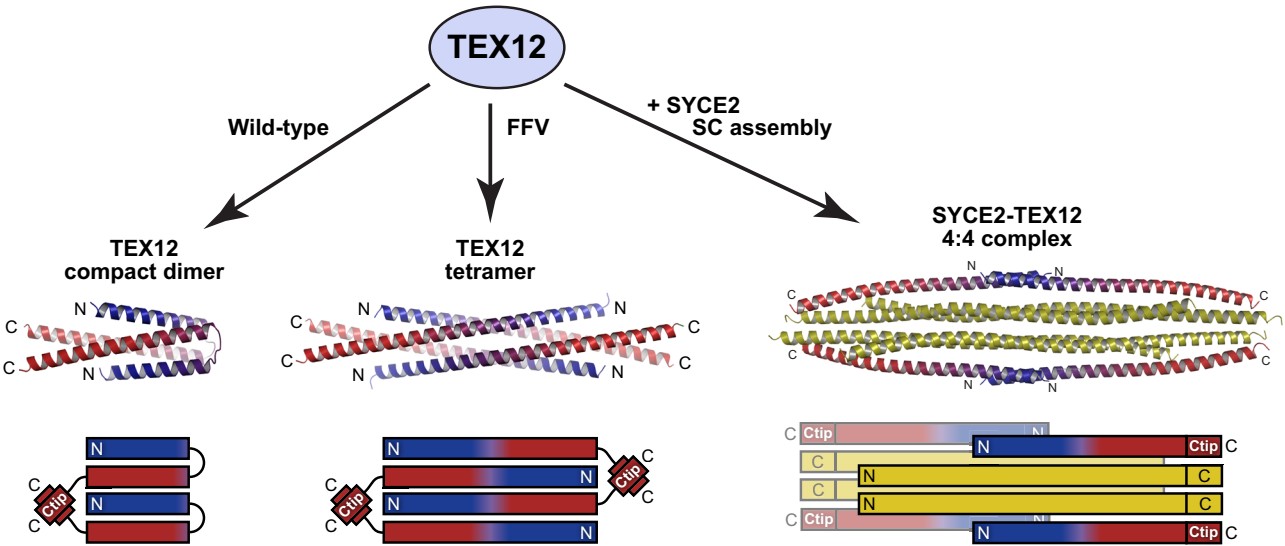

**Fig. 9 Structural diversity of TEX12 assemblies.** TEX12 adopts a compact dimer structure that undergoes conformational change into an extended tetramer upon FFV mutation. These structures share a common fold in which TEX12 chains interact in a four-helical bundle, stabilised by coiled-coil interactions between their Ctip sites. TEX12 chains adopt a helix–loop–helix conformation within the dimer, and open into linear helices within the tetramer. TEX12 also interacts with SYCE2 to form a 4:4 complex that assembles into fibres within the SC. Within this complex, TEX12 chains adopt linear conformations and orient Ctip sites to mediate assembly by forming four-helical structures with SYCE2's C-termini. The TEX12 compact dimer, TEX12 tetramer and SYCE2-TEX12 4:4 complex models are shown along with their schematics; TEX12 chains are coloured blue to red in an N- to C-terminal direction and SYCE2 chains are shown in yellow.

and Molecular Biosciences, Newcastle University). CD spectra were recorded in 10 mM $Na_2HPO_4/NaH_2PO_4$ pH 7.5, 150 mM NaF, at protein concentrations between 0.1 and 0.5 mg/ml, using a 0.2 mm pathlength quartz cuvette (Hellma), at 0.2 nm intervals between 260 and 185 nm at 4 °C. Spectra were averaged across nine accumulations, corrected for buffer signal, smoothed and converted to mean residue ellipticity ([θ]) (x1000 deg.cm$^2$.dmol$^{-1}$.residue$^{-1}$). Deconvolution was performed using the CDSSTR algorithm of the Dichroweb server (http://dichroweb.cryst.bbk.ac.uk)[34,35]. CD thermal denaturation was performed in 20 mM Tris pH 8.0, 150 mM KCl, 2 mM DTT, at protein concentrations between 0.1 and 0.4 mg/ml, using a 1 mm pathlength quartz cuvette (Hellma). Data were recorded at 222 nm, between 5 °C and 95 °C, at 0.5 °C intervals with ramping rate of 2 °C per minute, and were converted to mean residue ellipticity ([θ$_{222}$]) and plotted as % unfolded ([θ]$_{222,x}$−[θ]$_{222,5}$)/([θ]$_{222,95}$−[θ]$_{222,5}$). Melting temperatures (Tm) were estimated as the points at which samples are 50% unfolded.

**Size-exclusion chromatography small-angle X-ray scattering (SEC-SAXS).** SEC-SAXS experiments were performed at beamline B21 of the Diamond Light Source synchrotron facility (Oxfordshire, UK). Protein samples at concentrations >5 mg/ml were loaded onto a Superdex™ 200 Increase 10/300 GL size exclusion chromatography column (GE Healthcare) in 20 mM Tris pH 8.0, 150 mM KCl at 0.5 ml/min using an Agilent 1200 HPLC system. The column outlet was fed into the experimental cell, and SAXS data were recorded at 12.4 keV, detector distance 4.014 m, in 3.0 s frames. Data were subtracted and averaged, and analysed for Guinier region $Rg$ and cross-sectional $Rg$ ($Rc$) using ScÅtter 3.0 (http://www.bioisis.net), and $P(r)$ distributions were fitted using *PRIMUS*[36]. Ab initio modelling was performed using *DAMMIF*[37], in which 30 independent runs were performed in P1, P2 or P22 symmetry and averaged. Crystal structures and models were docked into *DAMFILT* molecular envelopes using *SUPCOMB*[38], and were fitted to experimental data using *CRYSOL*[39].

**Crystallisation and structure solution of TEX12 ΔCtip (6HK8).** TEX12 ΔCtip (49–113) protein crystals were obtained through vapour diffusion in hanging drops, by mixing 1.5 μl of protein at 60 mg/ml with 0.5 μl of crystallisation solution (25% (v/v) 1,4-dioxane) and equilibrating at 20 °C for 4 h. Crystals were promptly soaked in crystallisation solution containing cryoprotectant 20% ethylene glycol and were flash frozen in liquid nitrogen. X-ray diffraction data were collected at 0.9796 Å, 100 K, as 3600 consecutive 0.10° frames of 0.010 s exposure at beamline I03 of the Diamond Light Source synchrotron facility (Oxfordshire, UK). Data were indexed and integrated in *XDS*[40], scaled and merged in *Aimless*[41] using *AutoPROC*[42]. Crystals belong to hexagonal spacegroup P6$_5$22 (cell dimensions $a = 47.97$ Å, $b = 47.97$ Å, $c = 210.98$ Å, $\alpha = 90°$, $\beta = 90°$, $\gamma = 120°$), with two TEX12 chains per asymmetric unit. Data were corrected for anisotropy using the UCLA diffraction anisotropy server (https://services.mbi.ucla.edu/anisoscale/)[43], imposing anisotropic limits of 2.2, 2.2 and 2.1 Å, with principal

components of 7.52, 7.52 and −15.03 Å$^2$. Structure solution was achieved using *AMPLE*[44] through molecular replacement of Quark ab initio model decoys[45], with auto-tracing and re-building in *SHELX E*[46] and *PHENIX Autobuild*[47]. The structure was completed through manual model building using *COOT*[48] and refinement using *PHENIX* refine[49], with the addition of five dioxane (1,4-diethylene dioxide) ligands. Refinement was performed using isotropic atomic displacement parameters with two TLS groups for chain A and three TLS groups for chain B. The structure was refined against anisotropy corrected 2.11 Å data to R and Rfree values of 0.2291 and 0.2580, respectively, with 100% of residues within the favoured regions of the Ramachandran plot, clashscore of 2.30 and overall MolProbity score of 1.01.

**Crystallisation and structure solution of TEX12 FFV in a dimeric conformation (6HK9).** Crystals of TEX12 (49–123) FFV were obtained through vapour diffusion in sitting drops, by mixing 100 nl of protein at 43 mg/ml with 100 nl of crystallisation solution (200 mM calcium acetate, 40% MPD) and equilibrating at 20 °C. Crystals grew overnight and were harvested on the subsequent day and were cryo-cooled in liquid nitrogen. X-ray diffraction data were collected at 0.9790 Å, 100 K, as 2000 consecutive 0.010° frames of 0.010 s exposure on a Pilatus 6M detector at beamline I24 of the Diamond Light Source synchrotron facility (Oxfordshire, UK). Data were indexed and integrated in *XDS*[40], scaled and merged in *Aimless*[41] using *AutoPROC*[42]. Crystals belong to orthorhombic spacegroup C222$_1$ (cell dimensions $a = 43.233$ Å, $b = 219.712$ Å, $c = 37.501$ Å, $\alpha = 90°$, $\beta = 90°$, $\gamma = 90°$), with two TEX12 chains per asymmetric unit. Structure solution was achieved using *AMPLE*[44] through molecular replacement of Quark ab initio model decoys[45], with auto-tracing in *SHELX E*[46]. Phase improvement was achieved through iterative re-building by *PHENIX*[47]. The structure was completed through manual model building using *COOT*[48] and refinement using *PHENIX* refine[49], with the addition of seven calcium ions (based on coordination geometry and anomalous difference map peaks), two acetate ligands and one 2-Methyl-2,4-pentanediol (MPD) ligand. Refinement was performed using anisotropic atomic displacement parameters with riding hydrogen atoms. The structure was refined against 1.45 Å data to R and Rfree values of 0.1795 and 0.2047, respectively, with 100% of residues within the favoured regions of the Ramachandran plot, clashscore of 2.62 and overall MolProbity score of 1.05.

**Crystallisation and structure solution of TEX12 FFV in a helical conformation (6R2F).** Crystals of TEX12 (49–123) FFV were obtained through vapour diffusion in hanging drops, by mixing 1 μl of protein at 15 mg/ml with 1 μl of crystallisation solution (0.2 M LiNO$_3$, 40% MPD) and equilibrating at 20 °C for 8 months. Crystals were cryo-cooled in liquid nitrogen. X-ray diffraction data were collected at 0.9786 Å, 100 K, as 2000 consecutive 0.10° frames of 0.010 s exposure on a Pilatus 6M detector at beamline I24 of the Diamond Light Source synchrotron facility (Oxfordshire, UK). Data were indexed, integrated in *XDS*[40], scaled in

*XSCALE*[50], and merged in *Aimless*[41]. Crystals belong to orthorhombic spacegroup $I2_12_12_1$ (cell dimensions $a = 59.86$ Å, $b = 104.51$ Å, $c = 127.51$ Å, $\alpha = 90°$, $\beta = 90°$, $\gamma = 90°$), with two TEX12 dimers in the asymmetric unit. Structure solution was achieved using *AMPLE*[44] through molecular replacement of Quark ab initio model decoys[45], with auto-tracing in *SHELX E*[46]. Model building was performed through iterative re-building by *PHENIX Autobuild*[47] and manual building in *COOT*[48], with the addition of MPD ligands. The structure was refined with *PHENIX refine*[49], using isotropic atomic displacement parameters with four TLS groups per chain. The structure was refined against data to 2.29 Å resolution, to $R$ and $R_{free}$ values of 0.2378 and 0.2621, respectively, with 100% of residues within the favoured regions of the Ramachandran plot (0 outliers), clashscore of 10.63 and overall MolProbity score of 1.55 (Chen et al.[51]).

**Structural modelling**. A homodimeric coiled-coil of the TEX12 C-terminus, including its Ctip sequence (amino-acids 100–123) was modelled by *CCbuilder* 2.0[52] and was docked onto the ΔCtip crystal structure (6HK8) using *PyMOL* Molecular Graphics System, Version 2.3.2 Schrödinger, LLC. Manual editing was performed in *COOT*[48] in which minor refinement of atomic positions was required to achieve a seamless connection between crystal structure and the modelled C-terminus. The resultant structure was subjected to multiple rounds of energy minimisation by *Rosetta Relax*[53] interspersed with idealisation by *PHENIX* geometry minimisation[47] to achieve a final model. The wild-type compact dimer was modelled by taking one-half of the tetramer model and building loops of amino-acids 76–80 between the helical termini of the chains within each dominant dimer interface using *Rosetta Loop Modeling*[54].

**Molecular dynamics**. Molecular dynamics (MD) simulations were performed using AMBER ff19SB and OPC forcefields[55] in OpenMM[56], run locally on NVIDIA GeForce RTX 3090 GPU cards through a Google Colab notebook that was modified from the "Making-it-rain" cloud-based MD notebook[57]. The TEX12 compact dimer model was placed in a water box 10 Å larger than the structure, and was neutralised at a KCl concentration of 150 mM, by AMBER tleap[55]. The structure was equilibrated for 200 ps, and then run for 1 μs at 310 K and 1 bar pressure, using periodic boundary conditions, with the Langevin Middle Integrator and MonteCarlo Barostat, with integration times of 2 fs. The run was repeated three times. Structures in which the sequence register was shifted by between 1 and 8 amino-acids were run as above, for 100 ns simulations. MD trajectories were analysed using pytraj[58,59].

**Alphafold2 multimer modelling**. Models of the TEX12 dimer were generated using a local installation of Alphafold2[23], through the multimer pipeline[22], using templates dated no later than 01/09/2019 to prevent the inclusion of solved TEX12 structures. Data were analysed using modules from the ColabFold notebook[60].

**Protein sequence and structure analysis**. Multiple sequence alignments were generated using *Jalview*[61], and molecular structure images were generated using the *PyMOL* Molecular Graphics System, Version 2.0.4 Schrödinger, LLC.

**Statistics and reproducibility**. All biochemical and biophysical experiments were repeated at least three times with separately prepared recombinant protein material. Molecular dynamics simulations were performed in triplicate by repeating every step of the simulation from the same structural model.

**Reporting summary**. Further information on research design is available in the Nature Research Reporting Summary linked to this article.

## Data availability

Crystallographic structure factors and atomic co-ordinates have been deposited in the Protein Data Bank (PDB) under accession numbers 6HK8, 6HK9 and 6R2F. SAXS experimental data and models have been deposited in the Small Angle Scattering Biological Data Bank (SASBDB) under accession numbers SASDNN5, SASDNP5, SASDNQ5, SASDNR5 and SASDNS5. MD analysis data are included in Supplementary Data 1.

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

## Acknowledgements

We thank Diamond Light Source and the staff of beamlines I03, I24 and B21 (proposals mx13587, mx18598, sm14435, sm15580, sm15897, sm15836 and sm21777). We thank A. Baslé and H. Waller for assistance with X-ray crystallographic and CD data collection, C. Wood for advice on molecular dynamics, and J. Biemans for assistance with AlphaFold2 modelling analysis. J.M.D. is supported by a postdoctoral fellowship from the Herchel Smith Fund, Cambridge, UK. O.R.D. is a Wellcome Senior Research Fellow (Grant Number 219413/Z/19/Z).

## Author contributions

J.M.D. crystallised TEX12 FFV in both conformations and performed biophysical experiments. L.J.S. crystallised TEX12 ΔCtip and collected biophysical data. O.R.D. solved the TEX12 crystal structures, performed molecular dynamics simulations, analysed data, designed experiments and wrote the manuscript. J.M.D. and O.R.D edited the manuscript.

## Competing interests

The authors declare no competing interests.
