## [Peer Review File · Communications Biology]

Reviewers' comments:

Reviewer #1 (Remarks to the Author):

“Coiled-coil structure of meiosis protein TEX12 and conformational regulation by its C-terminal tip”

TEX12 is a component of the central element of the mammalian synaptonemal complex (SC). TEX12 was recently shown by the Davies laboratory to form a complex with SYCE2 that forms intermediate filament like fibres that facilitate the growth of the SC along the axes of homologous chromosomes. Recently it was also shown (by the Hunter, Davies and McClurg laboratories) that TEX12 localises to centrosomes during meiosis, but also during cancers, in a manner that is apparently independent of SYCE2. It thus became apparent that studying TEX12 in detail, without any of its binding partners, would yield information about how it might function outside of the SC, and how assembly of the SC might be regulated.

By combining X-ray crystallography with small-angle X-ray scattering (SAXS), Duncie et al provide a detailed study of the dimeric and tetrameric states of TEX12. The paper is well written, the figures beautifully presented, and the conclusions entirely in line with the data presented. I believe that the paper could be published as it is, but nonetheless I have a few minor points that the authors might want to consider including.

- 1) In the post-AlphaFold2 era it might be worthwhile reminding the reader why experimental structural studies are nonetheless essential. It is my understanding that, for example, such coiled-coil proteins as TEX12 are currently poorly amenable to AF2 predictions.
- 2) In my view Supplementary Figure 1 (at least the FFV profile at 1 mg/mL) belongs in the main Figure 2. It could be superimposed on Figure 1d, thus showing clearly that, unlike the Δ Ctip and LFIL mutants, FFV is a tetramer at both 1 and 10 mg/mL.
- 3) The compact dimer model of TEX12 is fascinating and may well provide an important insight into the regulation of SC formation (and quite possibly the role of TEX12 in centrosomes). Nonetheless the model would benefit from a little more supporting information. In the discussions the authors quite rightly refer to the dazzling array of different coiled-coil assemblies, and how these can change in response to, for example, the cellular environment. However, could the authors show the reader an example of a similar type of mechanism? That is to say a comparative figure of something similar (a hinge folding back or a transition from a tetramer to compact dimer). An example of a similar (ideally very similar) mechanism from elsewhere in biology would make the dimer model very convincing. I think the authors can dare to be speculative.
- 4) Again with the compact dimer model, I'm a little confused about the insertion of the loop, and how and where exactly this was introduced to create the compact dimer. Could the authors perhaps add an extra figure somewhere to show how exactly this was introduced? Likewise I think it would be good to highlight that the speculation is that there is a structured/unstructured conversion going on in the interface/six-amino acid interface/hinge/loop region. Maybe giving this region a single name throughout the manuscript would be helpful?
- 5) Further to the points above, could the authors perhaps highlight which residues would be inaccessible to the solvent in a compact dimer, if appropriate perhaps also showing the level of conservation of these residues (only if it's meaningful of course).
- 6) Figure 6 - (d) is labelled twice in the text.
- 7) In Figure 2, colouring the Δ Ctip and LFIL text in e) and f) in yellow and green (as in (g)) would help the reader navigate a little better

Reviewer #2 (Remarks to the Author):

Dunce et al, present a manuscript focusing on the structural and biophysical analysis of the TEX12 meiosis protein. TEX12 in association with SYCE2 play a role in synaptonemal complex assembly and function, which mediates homologous chromosome synapsis; TEX12 also play a role, independent from SYCE2 at centrosomes. To investigate structure and oligomerization properties of the TEX12 protein in the absence of SYCE2, thereby giving some functional insights on the role of TEX12 at centrosomes, they perform an elegant biophysical and structural analysis of TEX12 involving structure based mutants, x-ray crystallography, SEC MALS, CD, and SAXS.

The authors find out that the structure of TEX12 oligomers changes drastically upon C-terminal tip mutations which are required for facilitating the structural analysis of this system. The authors come up with a single unique structural state of the TEX12 oligomer which is consistent with the shape of a TEX12 oligomer as determined from in solution SEC-SAXS (compact dimer model). The author propose that this alternative oligomeric state, distinct to the one assumed in the context of the SC, may organise a storage pool of TEX12 at centrosomes, which will then be used for SC formation in meiosis.

This model is quite fascinating, therefore I would consider this work of interest for the Communications Biology readership. The manuscript is also well written and figures are clear and nice to look at. I would have one major concern, with the hope of improving the quality of this work even further before publication.

The authors present a collection of high resolution x-ray crystal structures of TEX12 C-terminal tip mutant oligomers, which is very interesting although none of these structures are consistent with the shape of wild type TEX12 assemblies as determined by in solution SAXS. They also have identified a mutation (F109E) that improves the behaviour of the protein and at the same time adopts the same shape of wild type TEX12 oligomers as determined by SAXS.

Would it be possible to further characterise the F109E mutant? Given its higher solubility can it be crystallised? Alternatively this sample could be used for NMR structural determination. In absence of a high resolution structure of this state (compact dimer) the conclusion about the TEX12 oligomer state remains a bit speculative and unsatisfactory.

Reviewer #3 (Remarks to the Author):

This manuscript determines X-ray crystal structures of the TEX12 protein in combination with their solution structures using X-ray scattering. TEX12 mutants demonstrate distinct dimer and tetramer conformations. Experiments are carefully executed and well documented. The model is supported by data presented. I have no serious concerns, but only have minor comments.

1. Page 9, "their cross-sectional radii were between 13-14 angstrom, suggesting that they are four-helical, rather than two helical, coiled-coils": This estimation is based on values determined in ref. 18. It will be helpful for readers to provide the basis for this estimation.

2. Trivial error: "in which" is duplicated on page 3, a little bit below the middle.

Response to reviewers' comments
'Coiled-coil structure of meiosis protein TEX12 and conformational regulation by its C-terminal tip'
(COMMSBIO-22-0707-T)

We are pleased that the reviewers recognise the value of this work, and are grateful for their thoughtful and considered comments. Here, we provide responses to the questions and comments raised by the reviewers, and outline how, in accordance with their requests, we have revised the manuscript.

Reviewer #1:

"Coiled-coil structure of meiosis protein TEX12 and conformational regulation by its C-terminal tip"

TEX12 is a component of the central element of the mammalian synaptonemal complex (SC). TEX12 was recently shown by the Davies laboratory to form a complex with SYCE2 that forms intermediate filament like fibres that facilitate the growth of the SC along the axes of homologous chromosomes. Recently it was also shown (by the Hunter, Davies and McClurg laboratories) that TEX12 localises to centrosomes during meiosis, but also during cancers, in a manner that is apparently independent of SYCE2. It thus became apparent that studying TEX12 in detail, without any of its binding partners, would yield information about how it might function outside of the SC, and how assembly of the SC might be regulated.

By combining X-ray crystallography with small-angle X-ray scattering (SAXS), Dunce et al provide a detailed study of the dimeric and tetrameric states of TEX12. The paper is well written, the figures beautifully presented, and the conclusions entirely in line with the data presented. I believe that the paper could be published as it is, but nonetheless I have a few minor points that the authors might want to consider including.

1) In the post-AlphaFold2 era it might be worthwhile reminding the reader why experimental structural studies are nonetheless essential. It is my understanding that, for example, such coiled-coil proteins as TEX12 are currently poorly amenable to AF2 predictions.

1. As the reviewer highlights, AlphaFold2 has limited usefulness for coiled-coil structure prediction (in our experience), which is likely owing to the subtle sequence differences between distinct coiled-coil structures, conformational changes, and the limited representation of coiled-coils (and importantly, those with correctly attributed oligomeric assemblies) in training datasets. We modelled TEX12 using AlphaFold2 multimer, excluding our solved structures from their use as templates. It predicted anti-parallel structures with coiled-coil interfaces that are similar to those present in the FFV helical structure (which does not appear to represent its solution structure and is likely a crystallographic artefact). However, it did not predict the biologically relevant dimer interface of the Δ Ctip and FFV dimer structures, or the helix-loop-helix structure of the wild-type compact dimer. In agreement with the reviewer, these observations highlight that structural studies remain essential for coiled-coil elucidation. We have added the

AlphaFold2 data in Supplementary Figure 8 and have summarised the above comments in the discussion (line numbers 340-348).

2) In my view Supplementary Figure 1 (at least the FFV profile at 1 mg/mL) belongs in the main Figure 2. It could be superimposed on Figure 1d, thus showing clearly that, unlike the Δ Ctip and LFIL mutants, FFV is a tetramer at both 1 and 10 mg/mL.

2. We agree and have superimposed the traces from supplementary figure 1 onto panel 2d, as suggested.

3) The compact dimer model of TEX12 is fascinating and may well provide an important insight into the regulation of SC formation (and quite possibly the role of TEX12 in centrosomes). Nonetheless the model would benefit from a little more supporting information. In the discussions the authors quite rightly refer to the dazzling array of different coiled-coil assemblies, and how these can change in response to, for example, the cellular environment. However, could the authors show the reader an example of a similar type of mechanism? That is to say a comparative figure of something similar (a hinge folding back or a transition from a tetramer to compact dimer). An example of a similar (ideally very similar) mechanism from elsewhere in biology would make the dimer model very convincing. I think the authors can dare to be speculative.

3. The most obvious similar mechanism is observed in cohesin (and bacterial MukBEF), in which their structures undergo conformational change from extended to folded structures through the induction of a looped structure towards the middle of their long coiled-coils that results in both ends of the coiled-coil becoming associated (Burmam *et al* 2019 *NSMB*, 10.1038/s41594-019-0196-z; Petela *et al* 2021 *eLife*, 10.7554/eLife.67268). We have added a description of this in the discussion (line numbers 361-364).

4) Again with the compact dimer model, I'm a little confused about the insertion of the loop, and how and where exactly this was introduced to create the compact dimer. Could the authors perhaps add an extra figure somewhere to show how exactly this was introduced? Likewise I think it would be good to highlight that the speculation is that there is a structured/unstructured conversion going on in the interface/six-amino acid interface/hinge/loop region. Maybe giving this region a single name throughout the manuscript would be helpful?

4. The location of the loop was directed by the internal symmetry of the FFV tetramer model (which replicates the Δ Ctip and FFV dimer interfaces). The R78-D82 salt bridges lie at the centre of symmetry of the molecule, so constitute the only location where it is possible to form loops that stabilise one half of the structure (i.e. the only location where an amino-acid on the left-right chain becomes proximal to the same amino-acid on its opposing right-left chain). We agree that this had not been adequately described, and its understanding is essential for the validity of the resultant model. Hence, we have included a figure showing the steps undertaken to produce the model (Supplementary Figure 3) and have added a thorough description in the results section (line numbers 240-253).

We also agree with the suggestion of giving this region a common name throughout, so have described this as a 'hinge' that is converted from an α -helical to a looped conformation upon compact dimer formation (Figures 3-8 and throughout the text).

5) *Further to the points above, could the authors perhaps highlight which residues would be inaccessible to the solvent in a compact dimer, if appropriate perhaps also showing the level of conservation of these residues (only if it's meaningful of course).*

5. The compact dimer model has the same hydrophobic core as the Δ Ctip and FFV tetramer, so is formed of almost entirely hydrophobic amino-acids. Further, the additional Ctip sequence forms a coiled-coil through its hydrophobic heptad amino-acids. Hence, the compact dimer structure buries almost entirely hydrophobic amino-acids, with charged/polar residues being solvent-exposed. We have added a description of this to the results section (line numbers 249-251) and have included a figure in which side-chains are coloured by hydrophobicity/charge (Supplementary Figure 4).

The conservation pattern of amino-acids shows that the most highly conserved amino-acids tend to be located at heptad positions of the Ctip and along the side of helices, whereas the least conserved amino-acids are solvent-exposed. However, most amino-acids have an intermediate conservation. This analysis is a little less informative than hydrophobicity, likely owing to the alternative conformation of TEX12 in its complex with SYCE2, so we have not included in the manuscript, but have included the figure below (coloured red=highly conserved, white=average and green=poorly conserved):

6) *Figure 6 - (d) is labelled twice in the text.*

6. We have removed the second label (line number 228).

7) *In Figure 2, colouring the ΔC_{tip} and LFIL text in e) and f) in yellow and green (as in (g)) would help the reader navigate a little better*

7. We agree and have changed the figure panels such that ΔC_{tip} and LFIL are coloured yellow and green, in keeping with the rest of the manuscript, and their dilutions are now in grey (Figure 2e,f).

Reviewer #2:

Remarks to the Author:

Dunce et al, present a manuscript focusing on the structural and biophysical analysis of the TEX12 meiosis protein. TEX12 in association with SYCE2 play a role in synaptonemal complex assembly and function, which mediates homologous chromosome synapsis; TEX12 also play a role, independent from SYCE2 at centrosomes. To investigate structure and oligomerization properties of the TEX12 protein in the absence of SYCE2, thereby giving some functional insights on the role of TEX12 at centrosomes, they perform an elegant biophysical and structural analysis of TEX12 involving structure based mutants, x-ray crystallography, SEC MALS, CD, and SAXS.

The authors find out that the structure of TEX12 oligomers changes drastically upon C-terminal tip mutations which are required for facilitating the structural analysis of this system. The authors come up with a single unique structural state of the TEX12 oligomer which is consistent with the shape of a TEX12 oligomer as determined from in solution SEC-SAXS (compact dimer model). The author propose that this alternative oligomeric state, distinct to the one assumed in the context of the SC, may organise a storage pool of TEX12 at centrosomes, which will then be used for SC formation in meiosis.

This model is quite fascinating, therefore I would consider this work of interest for the Communications Biology readership. The manuscript is also well written and figures are clear and nice to look at. I would have one major concern, with the hope of improving the quality of this work even further before publication.

The authors present a collection of high resolution x-ray crystal structures of TEX12 C-terminal tip mutant oligomers, which is very interesting although none of these structures are consistent with the shape of wild type TEX12 assemblies as determined by in solution SAXS. They also have identified a mutation (F109E) that improves the behaviour of the protein and at the same time adopts the same shape of wild type TEX12 oligomers as determined by SAXS.

Would it be possible to further characterise the F109E mutant? Given its higher solubility can it be crystallised? Alternatively this sample could be used for NMR structural determination. In absence of a high resolution structure of this state (compact dimer) the conclusion about the TEX12 oligomer state remains a bit speculative and unsatisfactory.

8. The reviewer correctly highlights that we have not crystallised the native conformation of TEX12, but have used high-resolution crystal structures of its alternative but related conformations, in combination with experimental SAXS and MALS data to build an atomic model of the native TEX12 compact dimer structure.

We agree with the reviewer the improved stability of the F109E mutant posed an excellent opportunity for experimental structure solution of its conformation, which appears to match that of the wild-type protein in solution. We had previously attempted its crystallisation unsuccessfully. Nevertheless, in light of the above comments, we re-explored this and spent the last four months attempting to obtain new diffraction-quality crystals. Unfortunately, and despite considerable experimental effort, we have not been

able to obtain useful crystals of this protein. In fact, we have rarely obtained any crystals as the protein remains soluble and undergoes phase transition in the majority of crystal conditions, even when crystallisation is attempted at its maximum obtainable concentration of 40 mg/ml and in different salt and pH conditions, and temperatures. The few crystals obtained were salt, non-diffracting (failed to optimise), tiny needles (far less than 1 μm across), or bundles of crystal shards that fragmented and dissolved during fishing. We have included example images below. The nature of the unusable crystals obtained is in keeping with our experience of other highly soluble coiled-coils that fail to crystallise. Overall, we have had to conclude that it is not currently technically possible to obtain a crystal structure of the F109E mutant.

Bundles of crystal shards

Tiny needles

Single crystal with diffraction pattern (5° combined slabs)

We had also considered the use of NMR, despite the known difficulties in using this technique for structure solution of oligomeric proteins. However, we found that protein expression was drastically reduced (to less than 10%) in minimal media required for labelling. As we were already using 12 litres of rich media to obtain protein of a final concentration of 40 mg/ml, it was not technically possible to increase the minimal media culture size to an amount necessary to obtain sufficient material. This problem is likely exacerbated by the need to express TEX12 as an MBP fusion protein, meaning that 4/5 of the protein mass expressed is the tag rather than the target protein. Despite many attempts at optimisation, we were not able to increase the yield to an amount that would enable labelling for NMR.

In light of the inability to obtain X-ray crystal or NMR structures of the wild-type (or F109E) compact dimer conformation, we explored an alternative approach of validating our structural model of the compact dimer by molecular dynamics simulations. We performed 1 μ s simulations of the wild-type compact dimer model at 37°C, atmospheric pressure, in triplicate, to assess whether the structure is stable and hence a plausible model. In these runs, the structure remained intact throughout the 1 μ s simulations, with retention of its hydrophobic core, Ctip coiled-coil, and their constituent amino-acid interactions, including R78-D82 salt bridges within the loops. The overall r.m.s. deviations remained within approximately 1.5-2.5 Å, with the lowest local r.m.s. fluctuations observed for the α -helices, including most of the Ctip (1 Å), slightly higher for the loops (1.5-2 Å), and much higher for the unstructured termini. We recognised the importance of providing controls to assess what would happen during molecular dynamics simulations of spurious structural models. Hence, we modified the structure by introducing sequence register shifts of between 1-8 amino-acids, and subjected each of these models to 100 ns simulations (in the same conditions as wild-type). In these cases, their cores ballooned almost immediately, giving overall r.m.s. deviations of between 4-6 Å within the first 10 ns. They then underwent further structural distortions during the full 100 ns runs. In some cases, this consisted of further ballooning apart of the core, and in others it consisted of complete distortion and dissociation of chains, giving overall r.m.s. deviations of up to 21 Å. Importantly, the differences between their behaviours can be explained by the 3.5 amino-acid periodicity within coiled-coils. Register shifts of 1, 2, 5, 6 and 8 gave the most dramatic distortions as hydrophobic amino-acids become surface exposed and polar amino-acids become buried. In contrast, register shifts of 3, 4 and 7 amino-acids gave less dramatic ballooning of the core as hydrophobic and polar were retained in buried and surface locations, with shifts of one or two helical turns, but were clearly unable to maintain the structure to the same level as the wild-type model. Hence, our molecular dynamics data demonstrate that the model we have presented is a plausible explanation of the compact dimer structure that is stable over long simulations and is readily disrupted by alternation of its primary amino-acid sequence. We have included these molecular dynamics data in the revised manuscript (Figure 7 and Supplementary Figures 5-6), and have described these data in the results section (line numbers 255-271).

Whilst we agree that it would be preferable to have a crystal structure of TEX12's wild-type conformation, we believe that we have provided the most thorough validation

possible of the model that we have generated from the conserved interfaces of its alternative conformations in combination with experimental SAXS and MALS data. Further, we believe that it is a strength of the manuscript that it provides a demonstration of how such ‘unsolvable’ structures may be predicted by adopting an integrative structural biology approach.

Reviewer #3:

Remarks to the Author:

This manuscript determines X-ray crystal structures of the TEX12 protein in combination with their solution structures using X-ray scattering. TEX12 mutants demonstrate distinct dimer and tetramer conformations. Experiments are carefully executed and well documented. The model is supported by data presented. I have no serious concerns, but only have minor comments.

1. Page 9, “their cross-sectional radii were between 13-14 angstrom, suggesting that they are four-helical, rather than two helical, coiled-coils”: This estimation is based on values determined in ref. 18. It will be helpful for readers to provide the basis for this estimation.

9. This is based on our experimental SAXS analyses of several other SC coiled-coil proteins that are dimers and tetramers, and in one case where the protein undergoes conformational change from dimer to tetramer upon changing the pH. In these previous studies, we found that dimers consistently give SAXS cross-sectional radii of between 8-9 Å, whereas tetramers give radii of 10-14 Å (Dunce *et al* 2018 *NSMB*, 10.1038/s41594-018-0078-9; Dunne *et al* 2019 *JBC*, 10.1074/jbc.RA119.008404; Sanchez-Sanchez *et al* 2020 *Sci Adv*, 10.1126/sciadv.abb1660; Dunce *et al* 2021 *NSMB*, 10.1038/s41594-021-00636-z). The latter is likely more variable as the inter-helical angle of tetramers can be more diverse than dimers, and hence give a wider range of widths. We have added a description of this and the additional references to the results section (line numbers 208-211).

2. Trivial error: “in which” is duplicated on page 3, a little bit below the middle.

10. We have removed this duplication.